# Increased dopaminergic neurotransmission results in ethanol dependent sedative behaviors in *Caenorhabditis elegans*

Pratima Pandey[1‡]*, Anuradha Singh[1‡], Harjot Kaur[2], Anindya Ghosh-Roy[2], Kavita Babu[1,3]*

**1** Department of Biological Sciences, Indian Institute of Science Education and Research (IISER) Mohali, Mohali, India, **2** National Brain Research Centre, Gurgaon, India, **3** Centre for Neuroscience, Indian Institute of Science (IISc), Bangalore, India

‡ These authors share first authorship on this work.
* pratima.sharma@babulab.org (PP); kavita.babu@babulab.org, kavitababu@iisc.ac.in (KB)

**Data Availability Statement:** All relevant data are within the manuscript and its Supporting Information files. The underlying numerical data for the graphs is attached as a supporting excel file.

## Abstract

Ethanol is a widely used drug, excessive consumption of which could lead to medical conditions with diverse symptoms. Ethanol abuse causes dysfunction of memory, attention, speech and locomotion across species. Dopamine signaling plays an essential role in ethanol dependent behaviors in animals ranging from *C. elegans* to humans. We devised an ethanol dependent assay in which mutants in the dopamine autoreceptor, *dop-2*, displayed a unique sedative locomotory behavior causing the animals to move in circles while dragging the posterior half of their body. Here, we identify the posterior dopaminergic sensory neuron as being essential to modulate this behavior. We further demonstrate that in *dop-2* mutants, ethanol exposure increases dopamine secretion and functions in a DVA interneuron dependent manner. DVA releases the neuropeptide NLP-12 that is known to function through cholinergic motor neurons and affect movement. Thus, DOP-2 modulates dopamine levels at the synapse and regulates alcohol induced movement through NLP-12.

## Author summary

We show that in the presence of ethanol, mutants in the D2-like dopamine autoreceptor, DOP-2 in *C. elegans* show a sedative phenotype. Our work goes on to reveal the mechanism of DOP-2 function in the presence of ethanol. Our initial analyses indicate that DOP-2 functions in the posterior PDE dopaminergic neuron to allow for normal locomotion in ethanol. We have also unearthed the mechanism of DOP-2 functioning and demonstrate that mutants in the *dop-2* autoreceptor show increased dopamine release, which in turn causes increased signaling from the neuron postsynaptic to the dopaminergic neuron PDE. This could in turn cause increased signaling at the cholinergic motor neurons, which results in increased body wall muscle contraction and leads to the locomotory defects seen in *dop-2* mutants treated with ethanol.

Since many movies were analysed in this work, we are happy to share the movies upon request. Only some of these movies are added as supplementary information. Request for movies may be sent to Mr. Salman Khan, IISc, whose email address is salmankhan@iisc.ac.in.

**Funding:** This work was supported by the Department of Biotechnology (DBT)-Wellcome Trust India Alliance (IA) fellowships [grant numbers IA/I/12/1/500516 and IA/S/19/2/504649] awarded to KB and partially supported by DBT (BT/PR24038/BRB/10/1693/2018), Ministry of Human Resources Development (MHRD)– Scheme for Transformational and Advanced Research in Sciences (STARS, STARS/APR2019/BS/454/FS) and Department of Science and Technology (DST)– Science and Engineering Research Board (SERB, SERB/F/7047) grants as well as a DBT-IISc partnership grant and IISER Mohali core funding to KB. PP is supported by a DST WOS-A grant [SR/WOS-A/LS-285/2018] and was earlier supported by a DBT Bio-CARe grant [BioCARe/01/10167]. AS was funded by the Council of Scientific and Industrial Research (CSIR)- University Grants Commission (UGC) for a graduate fellowship. AG-R lab is supported by the NBRC core fund from the Department of Biotechnology and a DBT/Wellcome Trust India Alliance fellowship [Grant number IA/I/13/1/500874] awarded to AG-R. The funders played no role in the study design, data collection and analysis, decision to publish, or preparation of the manuscript.

**Competing interests:** The authors have declared that no competing interests exist.

## Introduction

Alcohol is an easily available drug of abuse used around the world. Since excessive alcohol intake is detrimental to human health, many studies have focused on understanding the mode of action and dependency of this drug. Behavioral responses to alcohol and susceptibility to alcohol use disorders (AUDs) vary since they are dependent upon environmental, physiological and genetic differences amongst individuals [1,2]. Hence, it is still unclear how alcohol functions to modulate various behaviors, making it important to identify and analyze target gene/s and molecular pathways which functions to modulate behavioral phenotype/s upon alcohol intake.

Alcohol function(s) through multiple chemicals, peptides and proteins including acetylcholine (ACh), GABA, glutamate, dopamine (DA), neuropeptide-Y related pathways and ligand gated channels (Reviewed in [3–5]). Ethanol (EtOH) intake has been shown to increase DA release which, in turn, induces the reward pathway and brings about disinhibition of behaviors (reviewed in [6]). The DA pathway is signals through two types of DA receptors, characterized as D1-like (excitatory) and D2-like (inhibitory) receptors based on their functions and sequence homology (reviewed in [7]). These receptors belong to the superfamily of GPCRs (G protein-coupled receptors) and function through heterotrimeric G-protein coupling (α β γ) upon ligand binding. D1-like receptors transmit signals through the Gαs-subunits to stimulate adenylate cyclase activity, whereas D2-like receptors are coupled to Gαi/o, which result in decreased levels of adenylate cyclase and subsequent downstream signalling that regulates the activity of multiple other pathways [8–10]. A special class of D2-like autoreceptors that are located on dopaminergic neurons have also been identified; these receptors are thought to regulate the release of DA (reviewed in [11]). In mammals, activation of these receptors lowers the excitability of DA neurons and modulates the release and transmission of dopamine [12–14]. Studies have reported a correlation between D2 receptors and alcohol consumption and development of alcoholism [15–18]. Moreover, studies have also shown that D2 receptor levels are enhanced in alcoholics and that attenuation in the ligand activation of D2 receptors drives craving and relapse of alcoholism [19,20].

*Caenorhabditis elegans* as a model organism is popular for its powerful genetic tools [21] and has been widely utilized for studying various aspects of neuroactive drugs [22–26]. *Caenorhabditis elegans* encounters alcohol in its natural habitat (rotten fruits) and hence could be expected to have evolutionarily developed neuronal circuitry allowing for alcohol sensitivity. The DA system is very compact in *C. elegans*, with merely eight DA neurons as compared to ~500,000 neurons in the ventral midbrain of humans ([27] and reviewed in [28]). *Caenorhabditis elegans* DA receptors, like their mammalian counterparts, also belong to two subfamilies D1-like and D2-like receptors [29,30]. Studies in *C. elegans* have reported that EtOH administration causes dose dependent decline in the locomotor activity with increasing levels of EtOH exposure, which is similar to the depressive effects of EtOH seen in other animal systems [24,25,31]. The internal dose of EtOH responsible for this behavior is similar to that in mammalian systems, indicating that there might exist similar molecular targets [32].

Utilizing *C. elegans*, we devised an EtOH dependent assay and screened for dopaminergic receptors and pathway mutants. Mutants in the D2-like autoreceptor, *dop-2,* displayed an aberrant locomotory phenotype when exposed to 400 mM EtOH. DOP-2 has previously been shown to participate in associative learning, copulation behaviors and regulation of lifespan [25,33–36]. We found that *dop-2* mutant animals slowly dragged their body in concentric circles in what we refer to as Ethanol Induced Sedative (EIS) behavior. Our experiments indicate that *dop-2* mutants show increased dopamine release in the presence of EtOH, which is responsible for the EIS phenotype seen in these animals. Previous work has implicated a circuit

through the PDE sensory neurons, the DVA interneuron and cholinergic motor neurons that allow for normal locomotion in *C. elegans* [37,38]. Our work builds on this previous circuit and goes on to elucidate that the same circuitry is hyperactivated in *dop-2* mutants in the presence of EtOH and this in turn results in EIS behavior.

## Results

### Ethanol exposure affects movement in *dop-2* mutants

In *C. elegans*, the DA pathway is widely known to modulate egg laying, defecation, basal slowing, habituation, and associative learning [34,39,40]. DA receptors, DOP-1 and DOP-3 function antagonistically to regulate signaling in acetylcholine motor neurons [41], while the DA receptor, DOP-2 is expressed presynaptically in all the dopaminergic neurons and has the potential to function as an autoreceptor. However, *dop-2* mutants do not show very obvious defects associated with dopamine signaling that can explain its autoreceptor function [42,43]. It is possible that under wild-type conditions loss of *dop-2* is compensated for by other regulatory mechanisms in the organism. Modulation of behavior and function through the dopaminergic system upon exposure to drugs of abuse such as ethanol (EtOH) has been previously established [44,45]. We speculated that exposure of the DA pathway mutants to EtOH in the absence of food might allow us to screen for possible behavioral defects in these mutants. Previous work has shown that wild type (WT) *C. elegans* show flattening of the body bends at 400 mM concentration of EtOH [31,46]. We observed a similar phenotype with WT animals as published previously where after a period of 120 minutes (min) these animals recovered and began to move in a manner that was similar to WT animals that had not been exposed to EtOH ([46] and Fig 1A–1E and S1 Movie). We went forward to screen mutants in the DA pathway for defects in locomotion at 120 min in the presence of 400 mM EtOH and found that mutants in the dopamine autoreceptor, *dop-2*, showed a unique behavior where the *C. elegans* kept moving in circles while dragging the posterior part of their body (Figs 1D and 1E and S1A–S1D and S2 Movie). Mutants of *dop-2* treated with EtOH showed readily observable circular tracks on plates left overnight (approximately 16 hours (hr)) unlike that seen in EtOH treated WT animals (Fig 1F). We did not find this readily observable plate phenotype in other mutants of the dopamine pathway. However, upon quantitation we found that *cat-2* and *dop-3* mutants showed an increase in the number and/or amplitude of body bends (S1A and S1B Fig). As a control, mutants in the BK potassium channel, *slo-1* were also tested as these mutants have been shown to remain highly resistant to EtOH [31]. Quantitation of the *slo-1* locomotory behavior on EtOH revealed a large increase in both their number and amplitude of body bends when compared to WT control animals (S1A and S1B Fig).

On further analyzing the movement of *dop-2* mutants on EtOH we observed that there was slowing of locomotion as measured by the number of body bends and a flattening of the body bends (amplitude of body bends) that was very pronounced in the posterior half of the animal (Fig 1D and 1E). We refer to this behavior as an EtOH Induced Sedative (EIS) behavior and have used this EIS paradigm to gain insight into the function of DOP-2 in *C. elegans*. We initially performed a timecourse study on EtOH to understand the behavior of *dop-2* mutants in comparison to WT animals. We found that in the early time points of exposure to EtOH the *dop-2* mutants behaved in a manner similar to WT animals. Both WT and *dop-2* mutants were completely paralyzed within 10–12 min of EtOH treatment as has been previously reported for WT animals ([46], Figs 1A and S2A–S2E). WT *C. elegans* started showing recovery from paralysis within 30 min whereas *dop-2* mutants did not show recovery from prolonged EtOH exposure till 120 min (Figs 1A and S2A–S2D). Our experiments further revealed that even after 120 min, the *dop-2* animals could not overcome the flattening effect on their body bends especially

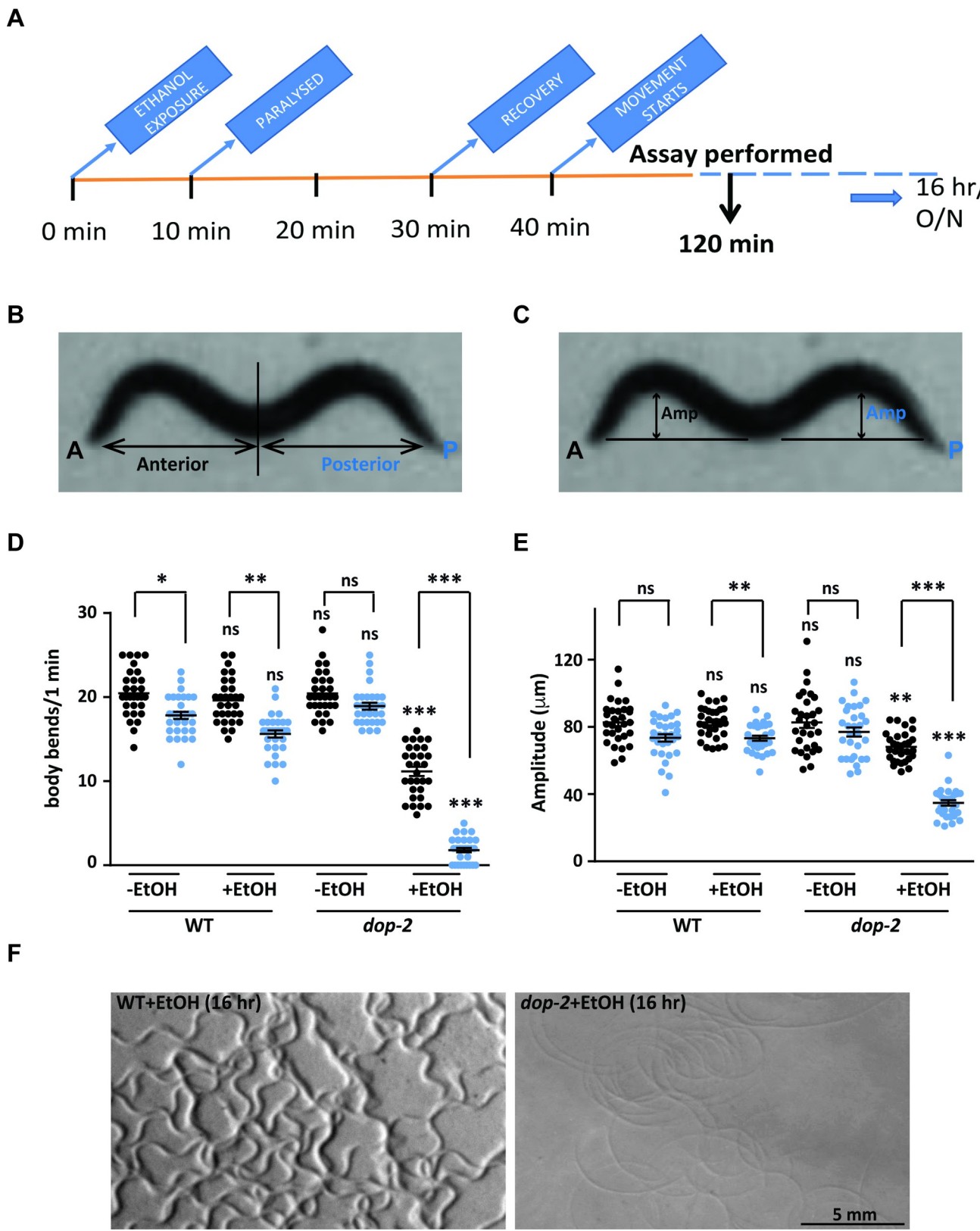

**Fig 1. Analysis of movement defects upon exposure to 400 mM ethanol.** (A) An illustration of the time-line for ethanol (EtOH) treatment in *C. elegans*. The time points showing paralysis and recovery of WT animals are indicated on the time-line. Experiments to test the number of body bends and

amplitude of body bends were performed at approximately the two-hour time point that is indicated in the time-line. (B) *C. elegans* body bends in the anterior (A) and posterior (P) regions marked by a partition. (C) Amplitude of body bends shown by a double-sided arrow indicating amplitude (Amp) in the anterior and posterior regions. (D) Quantitative analysis of number of body-bends for WT and *dop-2* mutant animals, both with and without EtOH treatment (n = 10, N = 3 and F = 201, DF = 7). (E) Quantitative analysis of amplitude of body-bends for WT and *dop-2* mutant animals, both with and without EtOH treatment (n = 10, N = 3 and F = 61.2 DF = 7). The same videos of moving animals were used to quantitate both number body bends and the amplitude of body bends for each genotype and treatment condition. Error bars represent ±S.E.M., "n" represents the number of animals and "N" represents the number of replicates. The p-values were calculated using one-way ANOVA and Tukey-Kramer multiple comparison test; "*" indicates p<0.05, "**" indicates p<0.01, "***" indicates p<0.001 and "ns" indicates not significant in all graphs. (F) Images of tracks from plates with WT and *dop-2* animals treated with EtOH and left overnight (16 hours (hr)) on the plate that was imaged. For all graphs the statistical comparison right above each genotype indicates a comparison with the respective WT control animals. Other statistical comparisons are indicated above lines indicating the genotypes that are being compared.

with respect to the posterior region of the animal's body (S2A–S2D Fig). In order to test if this behavior was seen only in the presence of EtOH or observable in untreated animals as well, we quantified the anterior and posterior body bends and amplitude of body bends from untreated WT and *dop-2* mutants and saw no significant difference between both the strains (Fig 1D and 1E). These data revealed that *dop-2* mutants showed locomotory defective movement upon EtOH exposure. Next, we wanted to test if loss of *dop-2* affect all muscles or largely affect the locomotory body wall muscle function. To address this question we decided to look at two sets of muscles; the egg-laying posterior muscles and the pharyngeal muscles. The egg-laying muscles are regulated by multiple neurotransmitters including dopamine [47], while pharyngeal pumping is regulated by serotonin (reviewed in [48]). However, studies have shown that serotonergic behaviors could be modulated by dopamine (reviewed in [49]). We found a significant decrease in egg-laying over 16 hr by *dop-2* mutants treated with EtOH when compared to WT control animals, while untreated WT and *dop-2* mutants showed no significant differences in egg-laying capacity (S3A and S3B Fig). We found no significant differences in pharyngeal muscle pumping of *dop-2* mutant animals treated with EtOH when compared with EtOH treated WT control animals (S3C Fig). These results suggest that *dop-2* mutants affect body wall muscles and not all muscles in the presence of EtOH.

### The EIS behavior in *dop-2* is modulated through the PDE neuron

Our results suggest that the EIS behavior occurs in the absence of *dop-2*. To show that it is indeed dependent on DOP-2 we made two independent transgenic rescue lines with the *dop-2p::dop-2::cfp* construct [35], and used these line in the EtOH assay and observed that both transgenic rescue lines could completely rescue the *dop-2* mutant phenotype (Fig 2A and 2B). We next wanted to identify the neuron through which DOP-2 could be functioning.

There are eight dopaminergic neurons in the *C. elegans* hermaphrodite, two pairs of CEP and a pair of ADE neurons in the head and a pair of PDE neurons in the posterior half of the *C. elegans* body [27]. All these dopaminergic neurons are mechanosensory in nature and control basal slowing behavior in the animal [39]. Our behavioral experiments indicated that the posterior half of the *dop-2* animals was more affected than the anterior in the presence of EtOH (Fig 1E and 1F). The only dopaminergic neuron with sensory endings at the posterior region is the PDE neuron that is also involved in harsh touch behavior and context dependent modulation of movement [38,50]. Bhattacharya et al., (2014) have shown that the PDE neuron through synaptic signaling via the DVA interneuron regulates the motor circuit in *C. elegans* [38,51]. From the above information, we hypothesized that the EIS behavior could be due to enhanced dopaminergic signaling from the PDE neuron. Since we were unable to find a PDE specific promoter for rescue experiments, we performed rescue experiments of the *dop-2* EIS phenotype using two separate promoters. The experiments were performed using the *gpa-14* promoter (expressed in a subset of neurons including the anterior dopaminergic ADE neurons

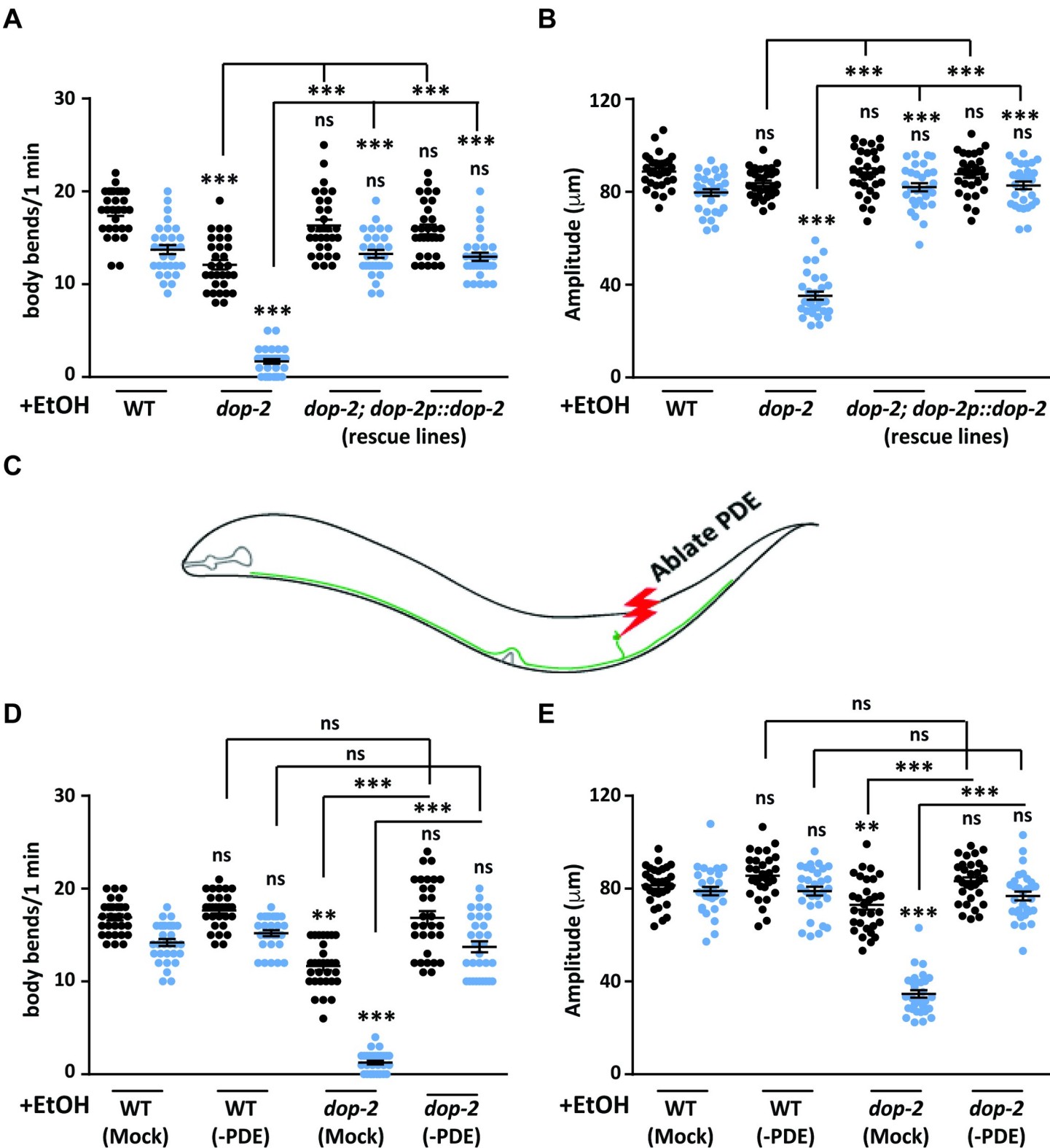

**Fig 2. DOP-2 functions in the PDE neuron for the Ethanol Induced Sedative (EIS) behavior.** (A) Graph indicating body bend measurements for rescue of the *dop-2* behavior using transgenic expression of *dop-2* under the *dop-2* promoter. Anterior body bends are shown in black and posterior body bends are shown in blue in all graphs (n = 10, N = 3 and F = 109, DF = 7). (B) Graph indicates amplitude of body bend measurements for WT, *dop-2* and rescue lines. Anterior amplitude of body bends are shown in black and posterior amplitude of body bends are shown in blue in all graphs (n = 10, N = 3 and F = 128, DF = 7). (C) Illustration of PDE neuron ablation. (D)

Quantitation of the number of body bends in mock ablated and PDE ablated animals, (n = 10, N = 3 and F = 140, DF = 7) (E) Quantitation of the amplitude of body bends in mock ablated and PDE ablated *C. elegans*, (n = 10, N = 3 and F = 84.5, DF = 7). All experiments in this figure were performed in the presence of EtOH (+EtOH). The same videos of moving animals were used to quantitate both number body bends and the amplitude of body bends for each genotype. Error bars represent ±S.E.M., "n" represents the number of animals and "N" represents the number of replicates. The p-values were calculated using one-way ANOVA and Tukey-Kramer multiple comparison test; "**" indicates p<0.01, "***" indicates p<0.001 and "ns" indicates not significant in all graphs. For all graphs the statistical comparison right above each genotype indicates a comparison with the respective WT control animals. Other statistical comparisons are indicated above or below lines indicating the genotypes that are being compared.

but not the posterior PDE neurons [52]) and the *gpa-16* promoter (expressed in a subset of neurons including the posterior PDE neurons but not the anterior dopaminergic neurons [52]). Assay results with these promoters indicated that *gpa-16* promoter (PDE expressing) could largely rescue the posterior body bends phenotype seen in *dop-2* mutants, while animals expressing *dop-2* under the *gpa-14* promoter in the *dop-2* mutant background were indistinguishable from *dop-2* mutant animals (S4A and S4B Fig). In order to get more insight into the possible function of *dop-2* in the PDE neuron, we decided to ablate the PDE sensory neurons in WT and *dop-2* backgrounds and analyzed the PDE ablated animals for EtOH sensitivity (Illustrated in Fig 2C). We observed that on exposure to EtOH the PDE ablated WT animals displayed no obvious defects in the number of body bends or amplitude of body bends when compared to the mock treated animals (Fig 2D and 2E and S3 and S4 Movies). However, upon testing the *dop-2* mutant animals in the EtOH assay we saw that unlike the mock ablated animals that showed the EIS behavior, the *dop-2* mutants with ablated PDE neurons behaved like control animals (Fig 2D and 2E and S5 and S6 Movies). These results indicate that the EIS behavior occurs in the absence of *dop-2* and that DOP-2 function in PDE neurons is sufficient to regulate this behavior. However, these results do not exclude the possibility that the DOP-2 neurons in the head could also be involved in this process.

## WT animals show EIS behavior in the presence of exogenous dopamine

Our data suggests that DOP-2 is functioning through the PDE neuron and could be modulating dopamine levels. D2 like autoreceptors have been shown to modulate the levels of dopamine through the regulation of transporters and components of the dopamine synthesis pathway (reviewed in [11]). However, the function of DOP-2 is still unclear. Since the EIS behavioral model provides us with an experimental system to investigate the function of DOP-2 in sedative movement during exposure to EtOH, we examined how DA levels and the DA synthesis pathway components might affect the EIS behavior. To address this we utilized the *cat-2* mutants in the EtOH assay. CAT-2 encodes a tyrosine hydroxylase and is required to synthesize DA from tyrosine. The *cat-2 (n4547)* mutant used in this study is an allelic deletion and is reported to have 20–30% of WT levels of dopamine [53]. We performed the EtOH assay with this mutant of *cat-2*. Upon observation and quantitation of *cat-2* mutant behavior it was quite evident that decreased dopamine levels suppressed the EIS phenotype (Fig 3A and 3B). We next generated *cat-2; dop-2* double mutants and performed the EtOH assay with these animals. We observed that this double mutant showed a similar behavior as was seen in *cat-2* mutant animals (Fig 3A and 3B). These results indicated that *cat-2* and *dop-2* function in the same pathway and that the EIS phenotype is not caused by decreased dopamine levels.

There are multiple reports indicating the negative regulatory role of the D2 autoreceptor in regulating synaptic levels of dopamine [54–56]. In order to test if the EIS phenotype was due to increased dopamine levels, we provided WT animals with exogenous dopamine in the EtOH assay. We observed that these animals showed the EIS behavior that was previously observed in *dop-2* mutants. These WT animals treated with exogenous dopamine showed

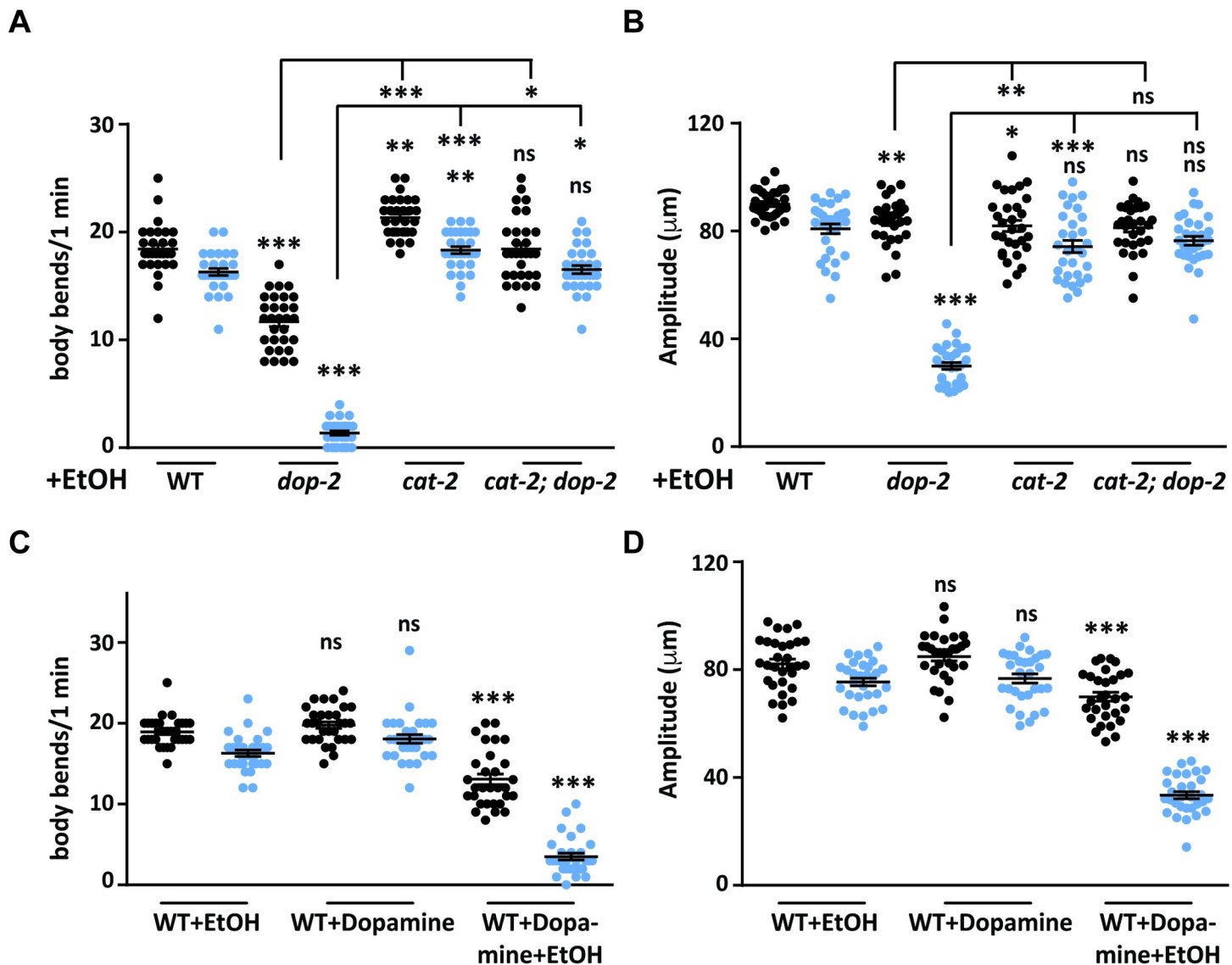

**Fig 3. Role of dopamine in the ethanol induced behavior.** (A) Graph represents the number of body bends in WT, *dop-2*, *cat-2 and cat-2; dop-2* mutants in the presence of EtOH. Anterior body bends are shown in black and posterior body bends are shown in blue in all graphs (n = 10, N = 3 and F = 265, DF = 7). (B) Quantitation of the amplitude of body bends in WT, *dop-2*, *cat-2 and cat-2; dop-2* strains in the presence of EtOH. Anterior amplitude of body bends are shown in black and posterior amplitude of body bends are shown in blue in all graphs (n = 10, N = 3 and F = 121, DF = 7). (C) Graph representing the number of body bends in WT animals under different conditions (+ EtOH, + DA and DA + EtOH), (n = 10, N = 3 and F = 169, DF = 5) (D) Graph shows the amplitude of body bends in WT *C. elegans* under different conditions (+ EtOH, + DA, + DA + EtOH), (n = 10, N = 3 and F = 141, DF = 5). The same videos of moving animals were used to quantitate both number body bends and the amplitude of body bends for each genotype. Error bars represent ±S.E.M., "n" represents the number of animals and "N" represents the number of replicates. The p-values were calculated using one-way ANOVA and Tukey-Kramer multiple comparison test; "*" indicates p<0.05, "**" indicates p<0.01, "***" indicates p<0.001 and "ns" indicates not significant in all graphs. For all graphs the statistical comparison right above each genotype indicates a comparison with the respective WT control animals. Other statistical comparisons are indicated below lines indicating the genotypes that are being compared.

decreased numbers and amplitude of body bends on EtOH plates. This phenotype was significantly different when compared to control WT animals treated with either just exogenous DA (no EtOH) or just EtOH (no exogenous DA) (Fig 3C and 3D and S7 and S8 Movies). These results suggest that increased dopamine release is responsible for the EIS behavior in *dop-2* mutant animals.

## Mutants in *dop-2* show increased dopamine release in the presence of EtOH

Our results so far indicate that there could be increased levels of dopamine release in *dop-2* mutants in the presence of EtOH, which is responsible for the EIS behavior in these mutants. To get more insight into the defects in dopamine release in the PDE neurons of *dop-2* mutant animals treated with EtOH, we used Fluorescence Recovery After Photobleaching (FRAP) recordings as a tool to examine DA release [57–59]. For these experiments we used a previously constructed strain with synaptobrevin-super ecliptic pHluorin reporter fusion (SNB-1:: SEpHluorin), which is expressed under the dopaminergic *asic-1* promoter ([60] and illustrated in S5 Fig). The PDE neuron synapses were examined with pH sensitive GFP (superecliptic pHluorin) attached to a vesicular protein SNB-1. The fluorescence was bleached at the synapses of the PDE neuron and the rate of recovery at the bleached area was calculated as a possible albeit indirect measure of DA release. Increased release of dopamine was monitored by the rate of recovery in PDE synapses post bleach. We observed that EtOH exposed *dop-2* mutants showed a significantly faster rate of recovery in the case of *dop-2* mutant animals in presence of EtOH (Fig 4A and 4B) and increased rates of fluorescence recovery at 60 s and 120 s time points in *dop-2* animals treated with EtOH (Fig 4C). We next wanted to test if *dop-2* mutants affect levels of the surface DA transporter DAT-1.

Previous reports indicate that D2 like receptors are involved in the cell surface localization of DAT-1 [61]. We utilized the previously constructed DAT-1::GFP translational fusion line to study DAT-1 expression in *dop-2* mutants [62]. We performed imaging and quantitated the cell surface expression of DAT-1 transporter in PDE neuron in the presence and absence of EtOH in the WT and *dop-2* mutant backgrounds. We observed a reduction in cell surface expression of DAT-1 in *dop-2* mutants exposed to EtOH (Fig 4D and 4E). These experiments indicate that *dop-2* mutants in the presence of EtOH show increased synaptic DA and decreased surface DAT-1.

DAT-1 is present on the DA neurons and recycles DA back into the neuron. We reasoned that if the reuptake mechanism was affected then *dat-1* (DA transporter) deletion mutant should also show EIS like phenotype since DA levels should be higher than WT, but that wasn't the case (S1A and S1B Fig). It is possible that loss of *dat-1* is only a part of the phenotype that allows for the EIS behavior in the animals that is seen in *dop-2* mutants treated with EtOH.

## DOP-2 functions through DOP-1 present in the DVA neuron

Thus far our data indicates that the EIS behavior seen in *dop-2* animals is modulated by the neurotransmitter DA. PDE has been found to be responsible for the DA effect but how this leads to defects in movement is still unknown. A previous study has elegantly shown that DA released from PDE neurons can activate DOP-1 present on the DVA neuron ([38] and reviewed in [51]). The DVA neuron upon activation results in the release of neuropeptide NLP-12 that results in the activation of downstream motor neurons [37]. The sensory neuron PDE forms strong synaptic connections with the interneuron DVA [39]. To understand the circuit through which PDE functions for the EIS phenotype we studied how the deletion of the dopaminergic receptor, *dop-1* could affect the movement in *C. elegans* after EtOH treatment. We found a small but significant decrease in body bends on comparing *dop-1* mutants with WT animals, however this decrease was less than the phenotype seen in *dop-2* mutant animals (Fig 5A and 5B). Next we made *dop-2; dop-1* double mutants and observed that it showed the same kind of behavior as *dop-1* mutants (Fig 5A and 5B). Thus, the *dop-1* deletion was able to suppress the EIS phenotype seen in *dop-2* mutants. If deletion of *dop-1* is obstructing DA signaling from PDE to DVA then we reasoned that expressing DOP-1 specifically in DVA would

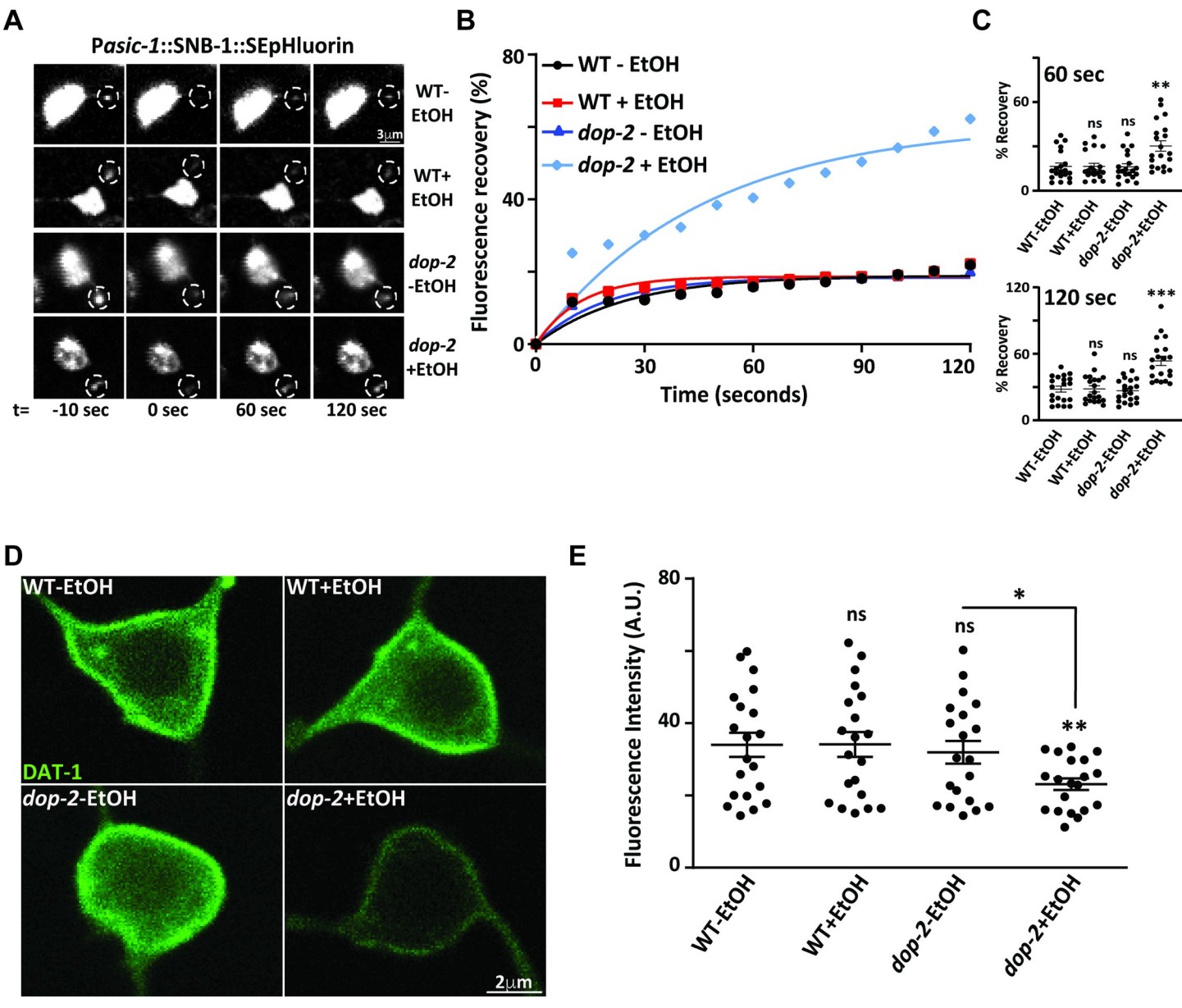

**Fig 4. Mutants in *dop-2* show increased dopamine release in presence of ethanol.** (A) Fluorescence recovery after photobleaching (FRAP) was performed on the PDE neuron synapses labeled with P*asic-1*::SNB-1::SEpHluorin. The images represent a PDE neuron synapse before bleaching (-10 seconds (sec)), followed by bleaching (0 sec) and post bleach at 60 sec and 120 sec. (B) Quantitation of rate of recovery taking the mean of all the recovery data points of each *C. elegans* in WT and *dop-2* mutant backgrounds with and without (+/- EtOH) treatment over 120 sec FRAP time course. Data represents 20–22 synapses per genotype. (C) Dot plots from FRAP data for percentage recovery were plotted for 60 sec (n = 10, N = 2 and F = 16.7, DF = 3) and 120 sec (n = 10, N = 2 and F = 20.3, DF = 3) time points respectively as we observed a sharp rise in percentage recovery at these time points. (D) Representative images of DAT-1::GFP expression in WT and *dop-2* mutant background with and without EtOH treatment. (E) Whole cell fluorescence quantification of DAT-1::GFP in PDE neurons for WT and *dop-2* mutants with and without EtOH, (n = 10, N = 2 and F = 3.09, DF = 3). Error bars represent ±S.E.M., "n" represents the number of animals and "N" represents the number of replicates. The p-values were calculated using one-way ANOVA and Tukey-Kramer multiple comparison test; "*" indicates p<0.05, "**" indicates p<0.01, "***" indicates p<0.001 and "ns" indicates not significant in all graphs.

restore the severe *dop-2* like EIS phenotype in the *dop-2; dop-1* double mutant animals. We indeed found that expressing DOP-1 specifically in DVA using the *nlp-12* promoter made the *dop-2; dop-1* animals revert to the *dop-2* like EIS behavior (Fig 5A and 5B). These findings indicate that EIS is dependent upon DA released from PDE and the DOP-1 receptor present on the DVA neuron.

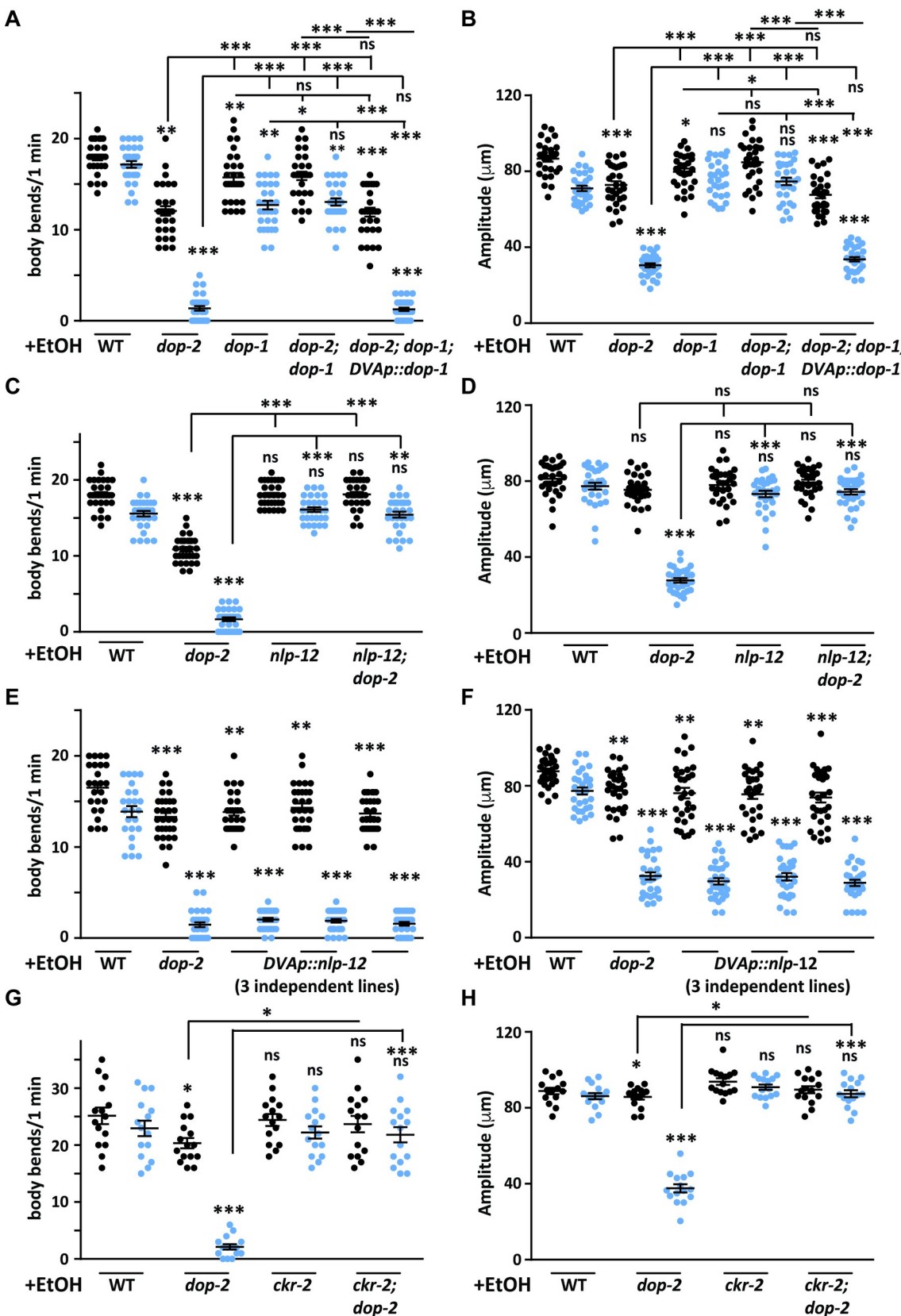

**Fig 5. DOP-2 functions through the DOP-1 and NLP-12 in the DVA neuron.** (A) Graph shows number of body bends in WT, *dop-2*, *dop-1*, *dop-2; dop-1* double mutants and the *dop-1* rescue line (*dop-2; dop-1; DVAp::dop-1*) on EtOH treatment. Anterior body bends are shown in black and posterior body bends are shown in blue in all graphs (n = 10, N = 3 and F = 212, DF = 9). (B) Graph shows amplitude of body bends in WT, *dop-2*, *dop-1*, *dop-2; dop-1* double mutant and the *dop-1* rescue line (*dop-2; dop-1; DVAp::dop-1*) on EtOH treatment. Anterior amplitude of body bends are shown in black and posterior amplitude of body bends are shown in blue in all graphs (n = 10, N = 3 and F = 105, DF = 9). (C) Quantitation of number of body bends in WT, *dop-2*, *nlp-12* and *nlp-12; dop-2* animals upon EtOH treatment, (n = 10, N = 3 and F = 295, DF = 7). (D) Quantitation of amplitude of body bends in WT, *dop-2*, *nlp-12* and *nlp-12; dop-2* animals upon EtOH treatment, (n = 10, N = 3 and F = 134, DF = 7). (E) Quantitation of number of body bends in WT, *dop-2* and three NLP-12 overexpression (OE) lines upon EtOH treatment, (n = 10, N = 3 and F = 302, DF = 9). The NLP-12(OE) lines do not show significant differences when compared to the *dop-2* mutant strain. (F) Quantitation of amplitude of body bends in WT, *dop-2* and three NLP-12(OE) lines upon EtOH treatment, (n = 10, N = 3 and F = 141, DF = 9). The NLP-12(OE) lines do not show significant differences when compared to the *dop-2* mutant strain. (G) Quantitation of number of body bends in WT, *dop-2*, *ckr-2* and *ckr-2; dop-2* animals upon EtOH treatment, (n = 5, N = 3 and F = 40.8, DF = 7). For this experiment we noticed that WT animals showed increased body bends when compared to WT data from other experiments. This change could be due to unavoidable difference in the experimental conditions as these experiments are carried out six months after the first set of experiment and the number of animals tested were also less than in other experiments. However similar to all other experiments, in these experiments too the difference between WT and *dop-2* mutant animals were obvious and easy to observe. (H) Quantitation of amplitude of body bends in WT, *dop-2*, *ckr-2* and *ckr-2; dop-2* animals upon EtOH treatment, (n = 5, N = 3 and F = 111, DF = 7). All experiments in this figure were performed in the presence of EtOH (+EtOH). The same videos of moving animals were used to quantitate both number body bends and the amplitude of body bends for each genotype. Error bars represent ±S.E.M., "n" represents the number of animals and "N" represents the number of replicates. The p-values were calculated using one-way ANOVA and Tukey-Kramer multiple comparison test; "*" indicates p<0.05, "**" indicates p<0.01, "***" indicates p<0.001 and "ns" indicates not significant in all graphs. For all graphs the statistical comparison right above each genotype indicates a comparison with the respective WT control animals. Other statistical comparisons are indicated above or below lines indicating the genotypes that are being compared.

The DVA neuron has been shown previously to function through the neuropeptide NLP-12. NLP-12 release potentially activates the downstream postsynaptic cholinergic motor neurons by binding to its receptors, CKR-2 (cholecystokinin like receptor) [37,63,64]. Interestingly, it has been shown previously that NLP-12 secretion is directly correlated with the speed of the animal [37]. We reasoned that loss of *nlp-12* in the *dop-2* mutant background might show a phenotype that is indifferent from that seen in *nlp-12* mutants if DOP-2 functions through NLP-12 to regulate the EIS behavior. Upon performing this experiment we found that *nlp-12* mutants behaved like WT animals in the presence of EtOH and the *nlp-12; dop-2* double mutants completely suppressed the *dop-2* phenotype that now behaved like WT and *nlp-12* mutant animals (Fig 5C and 5D).

We next hypothesized that increased NLP-12 secretion could allow the animal to show the EIS phenotype. To perform this experiment we overexpressed NLP-12 in the DVA neuron. We found that all three NLP-12 overexpression (OE) lines showed the EIS behavioral phenotype similar to that seen in *dop-2* mutants (Fig 5E and 5F and S9 Movie). Previous studies have shown that loss of *nlp-12* results in decreased depth of body bends while NLP-12(OE) lines show increased body bend depth in the absence of EtOH [38,65–68]. In order to ascertain these contrasting phenotypes shown by overexpression of NLP-12 in the presence and absence of EtOH, we first tested WT and the NLP-12(OE) strains without EtOH and food to ascertain baseline controls. WT *C. elegans* show an immediate change in their locomotory pattern when removed from food i.e. increased turning frequency for local search, but on prolonged deprivation of food their behavior changes to long runs and dispersal resulting in decreased amplitude [38, 69]. We tested a NLP-12(OE) line and found that similar to what was seen previously, the NLP-12(OE) line showed an increase in amplitude of body bends within 30 min off food ([38,68] and S6A Fig). As our experimental time line for EtOH exposure behavior is 120 min, we observed and quantitated the amplitude at this time point as well and observed a very substantial increase in the amplitude of body bends at the 120 min time point (S6A Fig). In contrast the *dop-2* mutants behaved like WT animals in the absence of EtOH (S6A Fig). However, when a similar experiment was performed in the presence of EtOH, both *dop-2* mutants and the NLP-12(OE) line behaved in a similar manner with decreased amplitude of

body bends (S6B Fig). Since NLP-12 functions in the DVA interneureuron, we went on to test if ablating DVA would affect the EIS behavior in the NLP-12(OE) line. However, we found that just ablating DVA in WT animals without EtOH treatment caused locomotory defects as has been shown previously ([70], S6C and S6D Fig and S10 Movie).

Previous studies have shown that NLP-12 peptides function through their receptor CKR-2 in cholinergic motor neurons [37,63,64]. In order to test if CKR-2 is involved in the EIS behavior seen in *dop-2* mutants, we tested both *ckr-2* and *ckr-2; dop-2* double mutants in the EtOH assay. Our data indicate that *ckr-2* mutants behaved in a manner that was similar to WT animals and the *ckr-2; dop-2* double mutants completely suppressed the *dop-2* mutant phenotype and showed a behavior similar to *ckr-2* and WT animals (Fig 5G and 5H). These data indicate that the increased dopamine seen in *dop-2* mutants, signals through DOP-1, increasing NLP-12 release, which in turn functions through CKR-2. Studies have shown that CKR-2 is expressed in motor neurons in the head and along the body of the animal [37,68]. Here we speculate that DOP-2 could be functioning through CKR-2 present on cholinergic motor neurons that are in proximity to the DVA neuron. In order to test if the EIS phenotype caused by NLP-12(OE) is also functioning through CKR-2, we made mutants of the NLP-12(OE) line with the *ckr-2* mutation and tested these animals for the EIS phenotype. Here again we found that *ckr-2* suppressed the EIS phenotype caused by NLP-12(OE) (S6E and S6F Fig).

In a recent study by Ramachandran *et al.* (2020), it has been suggested that NLP-12 also functions through the CKR-1 receptor. Hence, we went on to test *ckr-1* mutants for suppression of the *dop-2* EIS phenotype. We found that *ckr-1* mutants did not show a phenotype upon exposure to EtOH for 2 hr and the *ckr-1; dop-2* double mutants behaved just like the *dop-2* single mutant animals (S6G and S6H Fig). These data suggest that *dop-2* is not functioning upstream of *ckr-1* for the EIS phenotype.

Our data thus far suggests that dopamine released from PDE signals through DOP-1 receptors in the DVA interneuron, which in turn releases either one or both NLP-12 peptides in this circuitry and is responsible for the EIS behavior in *dop-2* mutants treated with EtOH.

## Increased acetylcholine levels at the NMJ results in EIS behavior

Previous work has shown that DA receptors DOP-1 (D1-like) and DOP-3 (D2-like) regulate locomotion in *C. elegans* [42,53]. Studies also indicate that the hypercontracted state observed in case of EtOH exposure in the animals is due to increased acetylcholine at the NMJ [25]. These studies prompted us to evaluate if dopamine might be involved in regulating locomotion through the cholinergic pathway. Initially we investigated if decreased levels of ACh could display the EIS behavior, but found no significant changes in the locomotion of cholinergic *cha-1* and *acr-16* mutants when compared to WT control animals on exposure to EtOH (S7A and S7B Fig). We next wanted to test animals with increased ACh release in the presence of EtOH. Previous work has shown that aldicarb treatment causes increased ACh signaling at the NMJ [37]. Hence, we performed the aldicarb assay followed by treatment of the animals with EtOH. For this experiment, *C. elegans* were exposed to 1 mM aldicarb followed by 400 mM EtOH exposure. We found that WT animals subjected to the above assay showed the EIS phenotype previously seen in *dop-2* mutant animals (Fig 6A and 6B and S11 Movie). The acetylcholine synthesis pathway mutant *cha-1* was used as a control as it is reported to show resistance to aldicarb [71]. These animals when exposed to aldicarb and EtOH did not show the defect in movement seen in WT *C. elegans* (Fig 6A and 6B). As a control *cha-1* mutants were also analyzed with just EtOH (no aldicarb) and these animals also behaved in a manner similar to that of WT control animals (S7A and S7B Fig). We also tested *dop-2* and *cha-1* mutants on normal growth plates without EtOH (-EtOH) and without aldicarb (-A) for any defects in body bends/

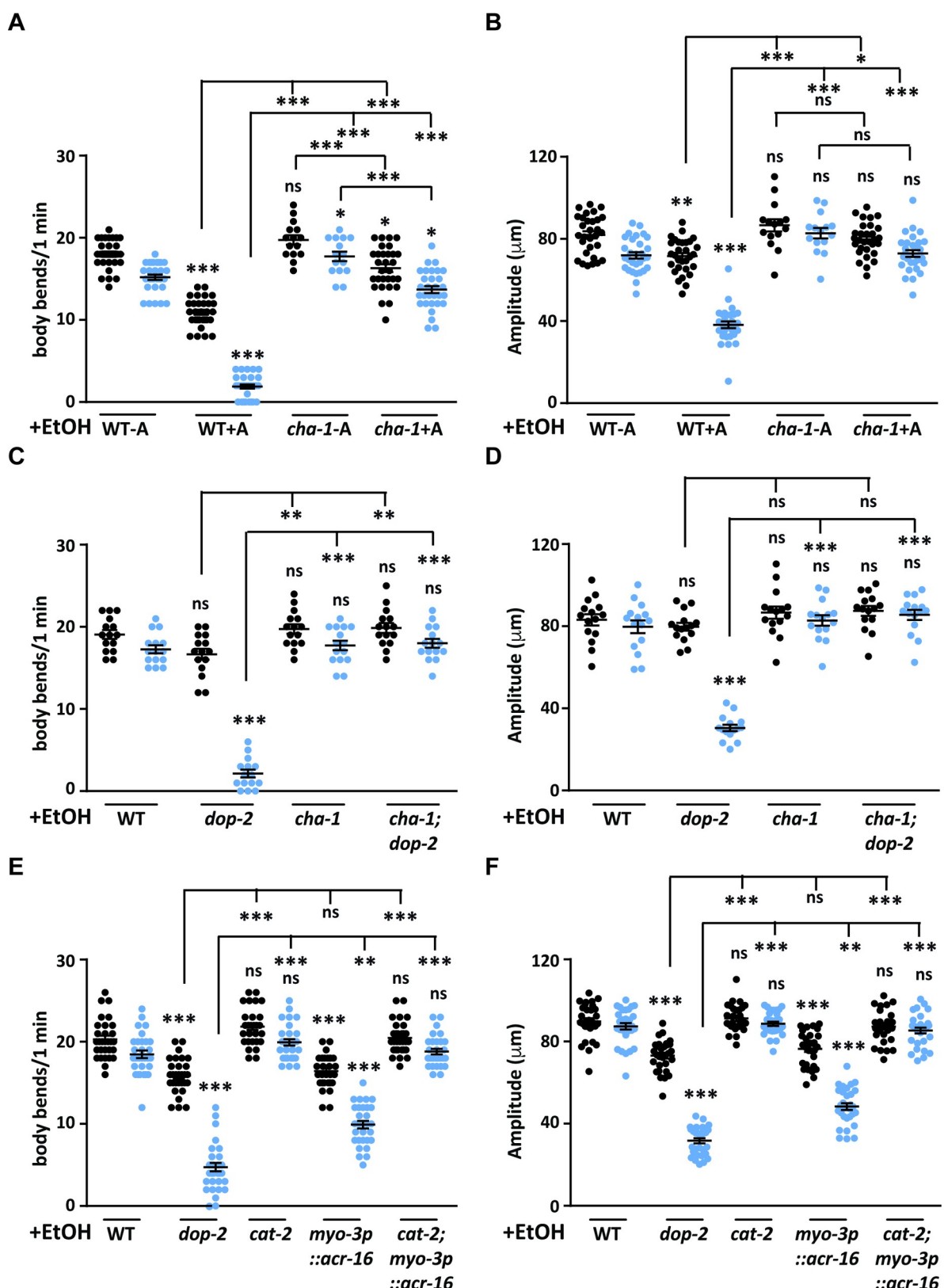

**Fig 6. Increased acetylcholine signaling causes increased sensitivity to ethanol in WT animals.** (A) Graph of number of body bends for WT *C. elegans* not treated with aldicarb (WT-A), WT animals treated with aldicarb (WT+A), *cha-1* mutants not treated with aldicarb (*cha-*

*1*-A) and *cha-1* mutants treated with aldicarb (*cha-1*+A) on EtOH plates. Anterior body bends are shown in black and posterior body bends are shown in blue in all graphs (n = 10, N = 3 and F = 209, DF = 7). (B) Graph indicates amplitude of body bends for WT *C. elegans* not treated with aldicarb (WT-A), WT animals treated with aldicarb (WT+A), *cha-1* mutants not treated with aldicarb (*cha-1*-A) and *cha-1* mutants treated with aldicarb (*cha-1*+A) on EtOH plates. Anterior amplitude of body bends are shown in black and posterior amplitude of body bends are shown in blue in all graphs (n = 10, N = 3 and F = 77.3, DF = 7). (C) Quantitation of number of body bends from WT, *dop-2*, *cha-1* and *cha-1; dop-2* lines on EtOH, (n = 5, N = 3 and F = 108, DF = 7). (D) Quantitation of amplitude of body bends from WT, *dop-2*, *cha-1* and *cha-1; dop-2* lines on EtOH, (n = 5, N = 3 and F = 57.6, DF = 7). (E) Quantitation of number of body bends from WT, *dop-2*, *cat-2*, *myo-3p::acr-16::gfp* line and the *cat-2; myo-3p::acr-16::gfp* lines on EtOH, (n = 10, N = 3 and F = 28.2, DF = 5). (F) Quantitation of amplitude of body bends from WT, *dop-2*, *cat-2*, *myo-3p::acr-16::gfp* and the *cat-2; myo-3p::acr-16::gfp* lines on EtOH, (n = 10, N = 3 and F = 32.4, DF = 5). All experiments in this figure were performed in the presence of EtOH (+EtOH). The same videos of moving animals were used to quantitate both number body bends and the amplitude of body bends for each genotype. Error bars represent ±S.E.M., "n" represents the number of animals and "N" represents the number of replicates. The p-values were calculated using one-way ANOVA and Tukey-Kramer multiple comparison test; "*" indicates p<0.05, "**" indicates p<0.01, "***" indicates p<0.001 and "ns" indicates not significant in all graphs. For all graphs the statistical comparison right above each genotype indicates a comparison with the respective WT control animals. Other statistical comparisons are indicated above or below lines indicating the genotypes that are being compared.

amplitude as compared to *C. elegans* exposed to aldicarb (+A). Data from our experiments indicated that *cha-1* mutants increased body bends in the presence of aldicarb, while *dop-2* mutants behaved like control animals (S7C and S7D Fig). In order to understand the role of cholinergic signaling in the EIS behavior we tested *dop-2*, *cha-1* and *dop-2; cha-1* mutants in the EtOH assay. We found that decreased ACh levels in *cha-1* mutants completely suppressed the *dop-2* EIS phenotype (Fig 6C and 6D).

These experiments along with our previous data implicate increased NLP-12 release in the EIS phenotype indicating that the *dop-2* EIS behavior could be an outcome of increased cholinergic signaling at the NMJ. ACh signals through the nicotinic ACR-16 receptors and the multiple levamisole-sensitive receptor subunits present on the postsynaptic body wall muscle membrane. In *acr-16* mutants there is an 85% decrease in response to ACh when compared to WT animals [72]. To understand the role of the acetylcholine receptors in the EIS behavior, we tested a previously used *myo-3p::acr-16::gfp* line for the EIS behavior [73]. We found that over-expression of ACR-16 caused the animals to show the EIS behavior while mutants in *acr-16* behaved like WT animals after EtOH treatment (Figs 6E and 6F and S7A and S7B). In order to ascertain that ACR-16(OE) is functioning through the dopaminergic pathway to cause the EIS phenotype, we analyzed the ACR-16(OE) line in the presence of the dopamine biosynthetic pathway mutant, *cat-2*. We found that the EIS phenotype of ACR-16(OE) was completely suppressed in the presence of *cat-2* (Fig 6E and 6F). These data indicate that ACR-16(OE) is functioning downstream of the dopaminergic pathway to give rise to the EIS phenotype. We also tested these ACR-16(OE) animals for defect in movement without EtOH and found that there is no significant defects in movement in this line in the presence or absence of aldicarb (S7C and S7D Fig).

Together, these data indicate that elevated muscle excitation through increased acetylcholine is playing a significant role in the EIS behavior of the animal.

## Discussion

DOP-2 belongs to a family of D2-like inhibitory receptors. It is thought to negatively regulate the release of dopamine by feedback inhibition of dopamine release from presynaptic neurons (Reviewed in [11,74,75]). In *C. elegans* DOP-2 is present on all the dopaminergic neurons as well as in the SIA interneuron, male spicules and ray neurons [35, 36]. However, the *dop-2* deletion does not appear to show similar defects in DA dependent behaviors as reported for other DA receptors such as DOP-1 and DOP-3 [39,41,42]. It is likely that behaviors associated with deletion of neuromodulatory molecules are not easily observable in native conditions since they are required to modulate multiple behaviors and not one specific behavior [11].

Ethanol (EtOH) has been shown to increase dopamine release from the mammalian ventral tegmental area and increased dopamine levels were found in the nucleus accumbens [76–78]. Since both loss of D2-like receptors and EtOH tend to increase DA levels, we tested *dop-2* deletion mutants for movement defect/s in the presence of EtOH and found a robust behavior involving decrease in body bends and the flattening of the body bends, which we have termed Ethanol Induced Sedative (EIS) behavior. This behavior was more pronounced in the posterior region of the animal. Although previous studies have reported that *C. elegans* show tolerance towards acute EtOH exposure and after recovery they exhibit various forms of disinhibitions in their behaviors [79,80], our studies describe for the first time the role of chronic EtOH treatment for extended periods of time (400 mM EtOH for up to 16 hr). Our data also shows that the EIS behavior in *dop-2* mutant animals is due to increased dopamine release.

The synaptic levels of dopamine are maintained by the activities of dopamine transporter (DAT-1), that recycles the dopamine back to the cell in conjunction with the DA autoreceptor (DOP-2) [56,81]. In mammals the activity of the DA transporter can be controlled by D2-autoreceptors by regulating their surface expression [56,82–86]. This occurs at least partially via an increase in DAT cell surface expression after D2-receptor activation [84]. However, D2- antagonists or D2 deletion could not alter DAT dependent DA uptake [54,86–90]. Our FRAP experiments conducted in the presence of increased EtOH showed increased release of DA in the absence of *dop-2*. A recent study by Formisano et al., (2020), also found that *dop-2* mutants result in accumulation of synaptic DA [59]. Further the membrane expression of DAT-1 was decreased in EtOH treated *dop-2* mutant animals when compared to control animals. Hence we show that DAT-1 surface transport could be modulated by DOP-2. However, since loss of *dat-1* does not show the EIS phenotype, it is likely that multiple factors including increased dopamine release contribute to the EIS behavior and it is not dependent on just loss of DAT-1 membrane expression.

All our data points towards the fact that the posterior region of the animal is more affected then the anterior region during EIS behavior in *dop-2* mutants. Neuronal ablation experiments demonstrate that the posterior DA neuron, PDE is responsible for the *dop-2* EIS phenotype. The PDE neuron forms multiple unidirectional synapses with the DVA interneuron. Our experiments implicate PDE function through the DVA neuron to allow for changes in downstream motor circuitry. These findings are in line with work that has previously reported that the PDE neuron makes direct synaptic contacts with the DVA neuron [38]. DVA is known to modulate locomotion both positively and negatively by providing a unique mechanism whereby a single neuron can fine-tune motor activity [70]. The DVA interneuron has connections with both motor neurons and interneurons and relays information for normal locomotion [38,69]. One mechanism of maintaining normal locomotion especially in conditions of stress like EtOH exposure could involve dopamine release from PDE regulating the movement of *C. elegans* through DVA. Our work further implicates the role of the DOP-1 receptor in DVA and the requirement of this pathway to maintain the EIS behavior seen in *dop-2* mutants. The DVA neuron expresses the DA receptor DOP-1, a D1-like excitatory receptor [38]. Further, in *D. melanogaster* and mammals it has been demonstrated that D1-like DA receptors promote EtOH- induced disinhibition [91,92].

A prior study has shown that movement induces NLP-12 release from DVA neurons and enhances ACh release at NMJs [37]. Further, multiple studies have demonstrated that neuropeptides modulate neuronal activity and synaptic transmission [37,93–99]. In this study we show that overexpressing just the NLP-12 neuropeptide in the DVA interneuron is sufficient to mimic the EIS phenotype seen in *dop-2* mutants, again implicating neuropeptides in synaptic functions.

Previous work that has shown that in *D. melanogaster*, mutations in a gene called *arouser* cause the animal to show increased sedation in the presence of EtOH as well as show increased boutons at the NMJ, indicating a link between increased neuromuscular signaling and greater susceptibility to alcohol [100]. Our data demonstrates that the EIS behavior seen in *dop-2* mutants could occur because of changes in neurotransmission at the *C. elegans* NMJ. We also show that increased neurotransmission brought about by the drug aldicarb or overexpressing acetycholine receptors at the body wall muscle both cause EIS behavior in *C. elegans*.

Overall our data implicates *dop-2* in the EtOH induced sedative phenotype by demonstrating that 1. Mutants in *dop-2* show increased dopamine release in the PDE neuron upon chronic EtOH treatment. 2. The increased dopamine released from the PDE sensory neuron may cause increased release of the neuropeptide, NLP-12 from the DVA interneuron. However, our data does not preclude the possibility that other dopaminergic neurons could also be involved in this process. These neurons may function through as yet unidentified mechanisms to allow for the EIS phenotype seen in *dop-2* mutant animals. 3. Increased signaling through NLP-12 could cause increased cholinergic neuronal function and result in the observed EIS phenotype. This circuit is illustrated in Fig 7.

Taken in its entirety, our work along with previous work paves a path for using *C. elegans* as a model system to study the molecular players involved in alcohol dependent locomotory functions [31,101].

## Methods

### Strains

Animals were maintained according to standard protocols [102]. N2 Bristol was used as the wild type (WT) strain. The mutant strains used in this study were; *dop-2(vs105)*, *dop-3(vs106)*, *dop-1(vs100)*, *cat-2(n4547)*, *cha-1(p1152)*, *acr-16(ok789)*, *slo-1(eg142)*, *dat-1(ok157)*, *nlp-12 (ok335)*, *ckr-1(ok2502)* and *ckr-2(tm3082)*. All strains used in this study have been backcrossed with WT *C. elegans* at least 2X times. Details of strains can be found in Table A in S1 Text.

### Constructs and transgenes

All constructs were generated using *pPD95.75* as the backbone with standard restriction digestion cloning procedures [103]. Transgenic lines were generated by microinjection of the desired plasmid as previously described [104]. The *dop-2p*::*dop-2*::*cfp* construct was obtained from Rene Garcia Lab [35]. The *nlp-12* promoter used in pBAB911 was cloned by amplifying a 355 bp upstream region of the *nlp-12* gene from genomic DNA using AACTGCAGGGCCG AGACGAATCCGGAGG (AS1) and CGGGATCCGCATTTTGTCGGAGGCAATT (AS4) primers and cloned into the *pPD95.75* vector using Pst I and Bam HI sites. The pBAB913 construct was generated by cloning *nlp-12p* into a previously made P*exp-1*::*sl2*::wrmScarlet vector using Pst I and Xma I sites that removed P*exp-1*. The pBAB912 construct contains a 1.7 kb genomic region of *nlp-12* amplified from genomic DNA using AACTGCAGGGCCGAGAC GAATCCGGAGG (AS1) and CGGGATCCGAAAATGTGTCGCTTCGAGAC (AS3) primers. The PCR amplified fragment was cloned into the *pPD95.75* vector using Pst I and Bam HI sites, for generating the *nlp-12* overexpression lines. For *dop-1* rescue experiment *dop-1* cDNA (1.2kb) was cloned under the *nlp-12* promoter in pBAB913 using Xma I and Kpn I sites. The pBAB916 and pBAB917 constructs contains *gpa-14 (*1.9 kb) and *gpa-16* (2.8 kb) promoters respectively amplified from genomic DNA and cloned upstream of the *dop-2* cDNA sequence amplified from the *dop-2p*::*dop-2*::*cfp* construct obtained from Rene Garcia Lab and cloned into the *pPD49.26* vector under NheI and XmaI sites. The promoters were cloned upstream using SphI and XmaI sites. To study the effect of DVA specific rescue or NLP-12

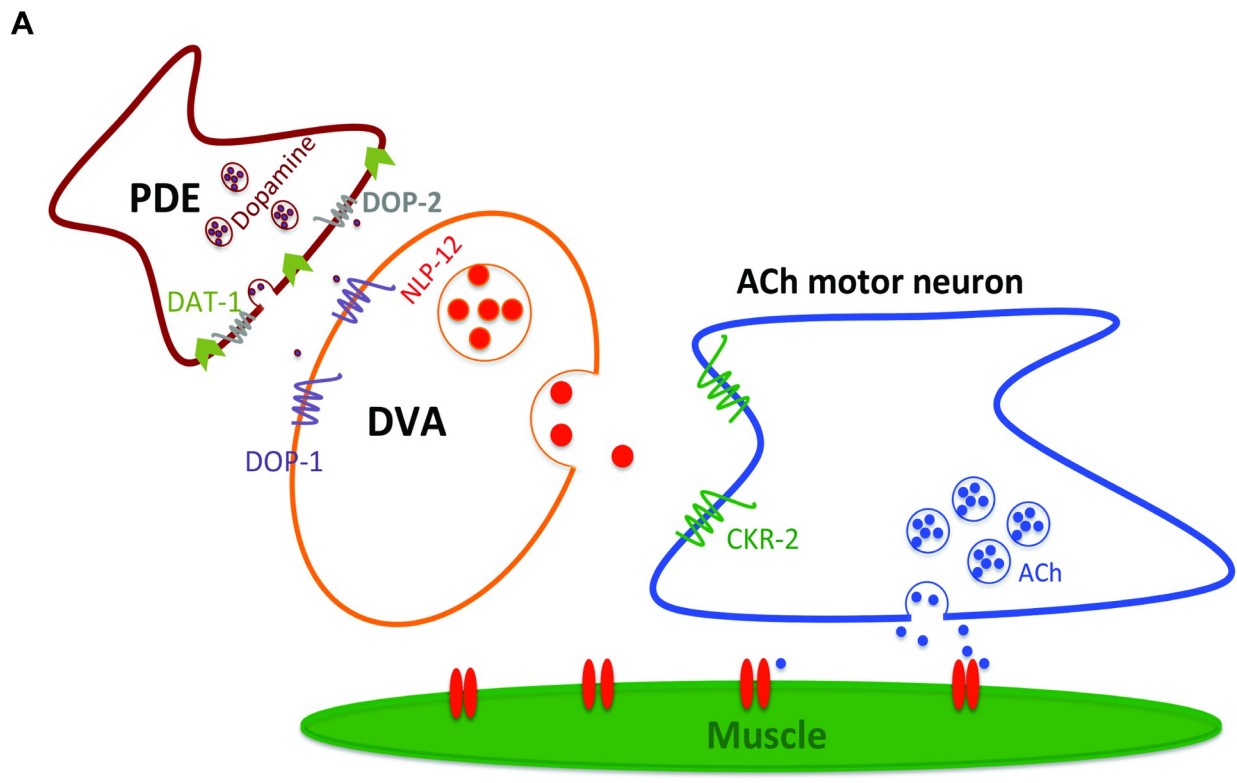

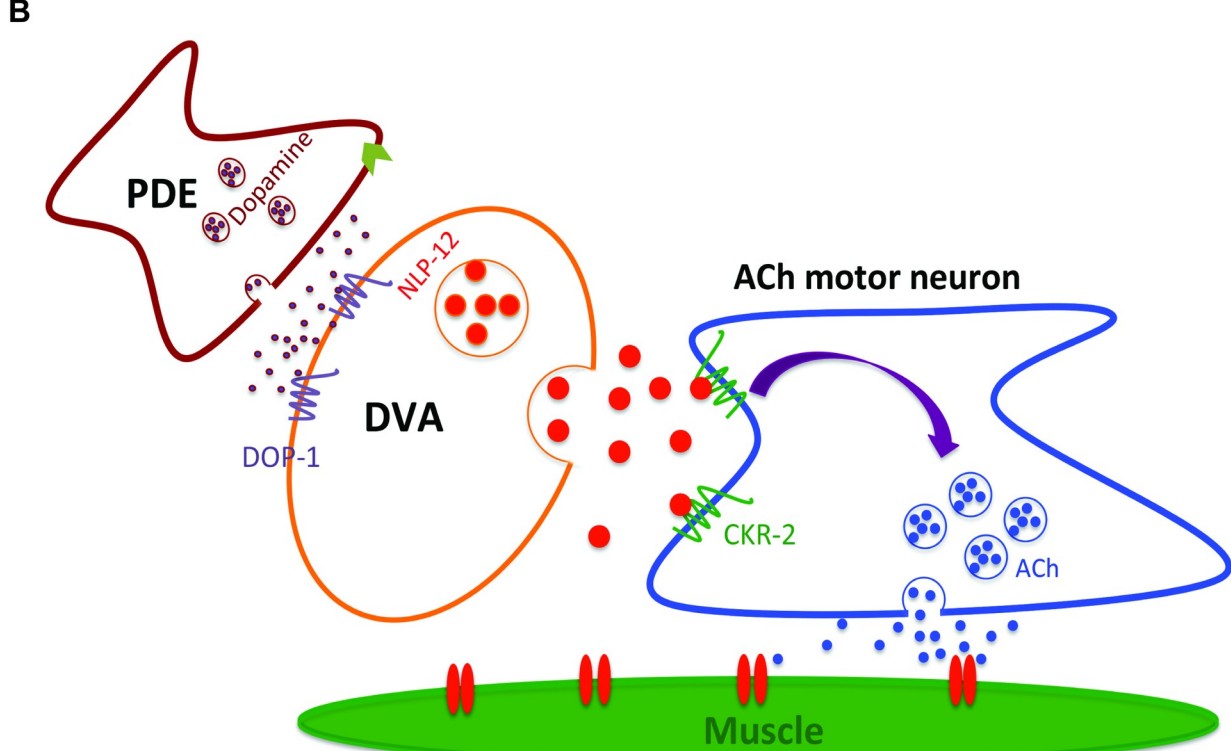

**Fig 7. Proposed model for DOP-2 functioning in the presence of EtOH.** (A) DOP-2 DA autoreceptors functions in the posterior DA neurons (PDE) to regulate DA levels, in the presence of EtOH. (B) Loss of the *dop-2* autoreceptor leads to unregulated release of DA in the presence of EtOH. The increased levels of DA activate the function of DOP-1 receptors present on the DVA neuron causing increased DVA activation. This

in turn may cause increased release of the neuropeptide NLP-12, which in turn could cause increased cholinergic signaling at the NMJ by binding to the CKR-2 receptor in motor neurons and activating the cholinergic motor neurons that are in close proximity to the DVA neuron. This model is based on this work and prior studies [37,38].

overexpression (OE), the *nlp-12* promoter was utilized to express DOP-1 or NLP-12 specifically in the DVA interneuron. The transgenic lines for these experiments are denoted as *DVAp::dop-1 and DVAp::nlp-12.*

A complete list of strains, primers and plasmids used in this study are available in the Tables A, B and C in S1 Text respectively.

## Ethanol induced behavioral assay

*C. elegans* were synchronized by bleaching and grown on nematode growth media (NGM) plates. They were maintained in well-fed conditions till the assay. The Ethanol (EtOH) plates were prepared using 60 mm unseeded plates i.e. without food plates that were dried for 3 hours under airflow in the biosafety cabinet. This was followed by spreading EtOH (400 mM) on the plates. The plates were then sealed with parafilm and were allowed to equilibrate for 120 minutes (min) at 20˚C. Well-fed adult animals (with 4–5 eggs) were initially transferred to unseeded plates for 15 to 20 seconds (sec) and then moved to the EtOH plates. 5 or 10 animals were used for each set and the experiment was done in triplicate for every genotype tested. All animals initially showed coiling behavior and paralyzed within 10 min of EtOH exposure as has been previously reported [31]. Recovery of *C. elegans* started in all the strains except *dop-2* in approximately 30 min. All the strains used for assays including WT showed Ethanol Induced Sedentary (EIS) -like behavior at around 30 min when movement started. However, most strains regained their normal locomotory behavior after more than 60 min. We performed a time dependent analysis of the locomotory behavior for both WT and *dop-2* mutant animals, to study the prolonged effect of EtOH. The time points included 10 min, 30 min, 60 min, 120 min, 300 min and overnight (16 hours (hr)) time points. We observed paralysis for both WT and *dop-2 C. elegans* within 10 min, however recovery was seen in WT but not *dop-2* mutant animals starting from the 30 min time point (S2 Fig). We also found that removing animals from EtOH after a 120 min treatment allowed *dop-2* mutant animals to recover within 60 min (S1E and S1F Fig). The quantitative analysis of behavior was done at 120 min of EtOH exposure for all the strains by recording the videos10 frames/sec for 1 min on the AxioCam MRm (Carl Zeiss) using the micromanager software. The anterior and posterior body bends and amplitude of body bends were quantified separately for each animal using the imageJ software [105]. The amplitude of body bends was calculated manually using the NIH ImageJ software with anterior and posterior body bends being quantitated separately. Quantitation of the amplitude was performed by measuring the distance between the deepest angle of the body bend and a tangential line from the tip of the head to the midsection of the body of the animal for the amplitude of the anterior body bend and from the midsection to the tail of the *C. elegans* for the amplitude of the posterior body bend, the measurements were normalized to the length of the animal to derive the values in micrometers. The number of body bends were scored manually using ImageJ keeping the same parameters in consideration for anterior and posterior body bends. Numbers of n = 15 or 30 animals were observed for each genotype for body bends and for the amplitude of body bends. Videos obtained from the same animal were used to quantify both body bends and amplitude of locomotion. The results were plotted as graphs using GraphPad Prism v6 and statistics were evaluated using one-way ANOVA.

## Exogenous dopamine assay

For exogenous dopamine application, 1 M freshly prepared dopamine was used as previously described [106]. The final concentration of dopamine used was 40 mM. Dopamine was spread on EtOH plates for EtOH + dopamine experiments and on unseeded dry plates for only dopamine exposure experiments. The plates were protected from light and used within 10 min of preparation for each assay. Animals were transferred from unseeded plates to the assay plate and their behavior was analyzed by making 1 min videos at 10 frames/sec on 120 min of dopamine/EtOH exposure.

## Aldicarb treatment prior to the EtOH assay

This assay was modified from the aldicarb assay to study the behavior of *C. elegans* on EtOH exposure after aldicarb treatment. The aldicarb assay was performed as described previously [107–109]. Briefly, plates with 1 mM aldicarb (Sigma- Aldrich 33386) were prepared 1 day prior to the assay and dried. *Caenorhabditis elegans* were transferred to aldicarb plates for 60 min and then moved to EtOH or no EtOH plates after which were identical except for the addition of EtOH. The animals were analyzed as previously described in the EtOH induced behavioral assay section.

## Microscopy

The DAT-1::GFP imaging experiment was performed on the Leica SP8 confocal microscope using the Argon laser at 10% gain. Young adult animals were immobilized using 30 mg/ml BDM (2,3-ButaneDione monoxime) on 2% agarose. All the image quantitation was done taking whole cell body expression of GFP using FIJI. Experiments were performed both with and without EtOH exposure.

## Neuronal ablation

The ablation of PDE and DVA neurons were done using Bruker Corporations ULTIMA two photon IR laser system. In which, one laser was used for imaging (920nm for GFP, 1040nm for mScarlet) and another laser for ablating the neurons (720 nm, irradiation of duration 20 ms, pulse width 80 fs, power~23 mW) as described in [110]. During ablation, L2 staged *C. elegans* were immobilized on 5% agarose pads using 10 mM levamisole hydrochloride (Sigma-Aldrich 10380000) or 0.1 μm-diameter polystyrene beads (00876–15; Polystyrene suspension). These animals were then recovered using mouth pipette onto the newly seeded NGM plates and were allowed to grow uptill young adult stage. The *C. elegans* were then evaluated in the EtOH induced behavioral assay.

## FRAP experiments

The increased extracellular release kinematics of dopamine in the presence of EtOH based on DA vesicle fusion, was analyzed and observed using the dopaminergic promoter (P*asic-1*) tagged with pH sensitive synaptobrevin-super ecliptic pHluorin reporter fusion construct (SNB-1::SEpHluorin) [34,57,60]. Young adult animals were mounted on 2% agarose pads and paralyzed using 0.05% levamisole hydrochloride (Sigma-Aldrich 10380000). FRAP experiments were performed on the Leica SP8 inverted confocal microscope. PDE synapses were identified by Synaptobrevin::SEpHluorin fluorescence. Bleaching was done using the 488 nm argon laser, 80% bleach power for 5–10 sec to an intensity of 10–15% of original fluorescence value. Fluorescence was monitored every 10 sec for 2 min and analyzed using FIJI software. The percentage recovery was calculated at each time point by dividing the final fluorescence

values by the initial fluorescent value after bleaching and 20 synapses were analyzed per genotype. The data was plotted using GraphPad Prism v6 and analyzed using non-linear regression plotting and one phase association exponential equation was used to analyze this data.

## Statistical analysis

All statistical analyses were performed by using GraphPad Prism Version 6.0. The error bars represent SEM. Statistical comparisons were done using one-way ANOVA with Tukey-Kramer multiple comparison test and Student's t-test with Welch's correction. The level of significance was set as "*" indicates $p < 0.05$, "**" indicates $p < 0.01$ and "***" indicates $p < 0.001$.

## Supporting information

**S1 Fig. Ethanol dependent phenotype of dopaminergic pathway mutants.** (A) Graph of number of body bends (anterior body bends are shown in black and posterior body bends are shown in blue in all graphs) in wild type (WT), *cat-2*, *dat-1*, *dop-1*, *dop-2*, *dop-3* and *slo-1* animals upon Ethanol (EtOH) treatment, (n = 10, N = 3 and F = 66.8, DF = 13). Please note that although we did not find a significant difference between WT and *dop-1* animals in this experiment, we did however find a significant decrease in the number of body bends seen in *dop-1* mutants that were outcrossed more times. These data are shown in Fig 5A. (B) Graph of amplitude of body bends (anterior amplitude of body bends are shown in black and posterior amplitude of body bends are shown in blue in all graphs) for WT, *cat-2*, *dat-1*, *dop-1*, *dop-2*, *dop-3* and *slo-1* mutants upon EtOH treatment, (n = 10, N = 3 and F = 82.2, DF = 13). (C) Graph of number of body bends plotted from pooled data of WT and *dop-2* animals upon EtOH treatment, (n = 30, N = 10 and F = 1196, DF = 3). (D) Graph of amplitude of body bends from pooled data of WT and *dop-2* animals upon EtOH treatment, (n = 30, N = 10 and F = 1920, DF = 3). Our data throughout this manuscript show differences in the values plotted for WT and *dop*-2 animals. In order to get a better understanding of all the data included throughout this manuscript, data from all experiments were pooled for WT and *dop-2* mutants and depicted as a bar-graph in figures C and D. (E) Graph of number of body bends quantitated for *dop-2* mutant animals under different conditions including 2 hours (hr) on EtOH (-food) and the same animals transferred to NGM plates with food to study the recovery of EIS behavior in *dop-2* mutant animals, observed and quantitated after 1 hr on food, (n = 10, N = 3 and F = 406, DF = 3). (F) Graph of amplitude of body bends quantitated for *dop-2* mutant animals under different conditions including 2 hr on EtOH (-food) and the same animals transferred to NGM plates with food to study the recovery of EIS behavior in *dop-2* mutant animals, observed and quantitated after 1hr on food, (n = 10, N = 3 and F = 226, DF = 3). The same videos of moving animals were used to quantitate both body bends and the amplitude of body bends for each genotype. Error bars represent ±S.E.M., "n" represents the number of animals and "N" represents the number of replicates. The p-values were calculated using one-way ANOVA and Tukey-Kramer multiple comparison test; "**" indicates p<0.01, "***" indicates p<0.001 and "ns" indicates not significant in all graphs. For all graphs the statistical comparison right above each genotype indicates a comparison with the respective WT control animals. (TIF)

**S2 Fig. Time-dependent ethanol assay.** (A) Number of anterior body bends quantified from WT and *dop-2* mutant upon EtOH treatment at 10 minutes (min), 30 min, 60 min, 120 min, 300 min and overnight (16 hr) time points (n = 10, N = 3 and F = 239, DF = 11). (B) Number of posterior body bends quantified from WT and *dop-2* mutant animals during an EtOH assay at 10 min, 30 min, 60 min, 120 min, 300 min and overnight (16 hr) time points (n = 10, N = 3

and F = 272, DF = 11). (C) Amplitude of anterior body bends quantified from WT and *dop-2* mutant animals upon EtOH treatment at 10 min, 30 min, 60 min, 120 min, 300 min and over-night (16 hr) time points (n = 10, N = 3 and F = 372, DF = 11). (D) Amplitude of posterior body bends quantified from WT and *dop-2* mutant animals upon EtOH treatment at 10 min, 30 min, 60 min, 120 min, 300 min and overnight (16 hr) time points (n = 10, N = 3 and F = 263, DF = 11). (E) Diagrammatic representation of the EtOH assay over time (this image is also shown in Fig 1A) and a representation of WT and *dop-2* EIS phenotypes on EtOH assay plates (bottom right). The same videos of moving animals were used to quantitate both number of body bends and the amplitude of body bends for each genotype. Error bars represent ±S.E.M., "n" represents the number of animals and "N" represents the number of replicates. The p-values were calculated using one-way ANOVA and Tukey-Kramer multiple comparison test; "**" indicates p<0.01, "***" indicates p<0.001 and "ns" indicates not significant in both graphs. For all graphs the statistical comparison right above each genotype indicates a comparison with the WT control animals at 10 min post EtOH treatment. Other statistical comparisons are indicated above lines indicating the genotypes that are being compared.
(TIF)

**S3 Fig. Mutants in *dop-2* show defects in egg-laying behavior.** (A) Graph depicting the number of eggs laid over a 16 hr time period for WT and *dop-2* animals. The assay was performed on plates with food and without EtOH. The data was plotted from 12 animals for both genotypes (t = 0.2030 and df = 21.04). (B) Graph depicting the number of eggs laid over a 16 hr time period from WT and *dop-2* animals. These assays were performed on plates without food and with EtOH. The data was plotted from 12 animals for both genotypes (t = 0.2030 and df = 21.04). (C) Number of pharyngeal pumps per min, recorded from WT and *dop-2* animals after 120 min of EtOH treatment. The data were plotted from 20 WT and 20 *dop-2* animals, (t = 0.26 and df = 34.05). Error bars represent ±S.E.M. and p-values were calculated using *t-test* with Welch's correction; "***" indicates p<0.001 and "ns" indicates not significant.
(TIF)

**S4 Fig. Rescue of EIS behavior in *dop-2* mutants using head (ADE) and tail (PDE) dopaminergic neuron expressing promoters.** (A) Graph representing the number of body bends quantitated from WT, *dop-2*, *dop-2*; *gpa-16p::dop-2* and *dop-2*; *gpa-14p::dop-2* lines. Anterior body bends are shown in black and posterior body bends are shown in blue in the graph (n = 10, N = 3 and F = 336, DF = 7). (B) Graph representing amplitude of body bends quantitated from WT, *dop-2*, *dop-2*; *gpa-16p::dop-2* and *dop-2*; *gpa-14p::dop-2* lines. Anterior amplitude of body bends are shown in black and posterior amplitude of body bends are shown in blue the graph (n = 10, N = 3 and F = 181, DF = 7). The same videos of moving animals were used to quantitate both number of body bends and the amplitude of body bends for each genotype. Error bars represent ±S.E.M., "n" represents the number of animals and "N" represents the number of replicates. The p-values were calculated using one-way ANOVA and Tukey-Kramer multiple comparison test; "**" indicates p<0.01, "***" indicates p<0.001 and "ns" indicates not significant in both graphs. For both graphs the statistical comparison right above each genotype indicates a comparison with the respective WT control animals. Other statistical comparisons are indicated above lines indicating comparisons with *dop-2* mutants.
(TIF)

**S5 Fig. Diagrammatic representation of the FRAP experiment.** A pH sensitive GFP Fluorophore, pHluorin, was tagged to the synaptic vesicle protein SNB-1 and expressed in DA neurons. *Caenorhabditis elegans* lines expressing the pHluorin construct were imaged to

quantitate dopamine release. This construct was previously used in [60].
(TIF)

**S6 Fig. DVA ablation affects locomotion in WT animals.** (A) Quantitation of the amplitude of body bends (anterior body bends are shown in black and posterior body bends are shown in blue in all graphs) from WT, *dop-2* mutant animals and an NLP-12 overexpression line where NLP-12 is expressed in DVA neurons. These experiments were performed at 30 and 120 min off food without EtOH treatment, (n = 10, N = 2 and F = 104, DF = 11). (B) Quantitation of the amplitude of body bends (anterior amplitude of body bends are shown in black and posterior amplitude of body bends are shown in blue in all graphs) from WT, *dop-2* mutant animals and an NLP-12 overexpression line where NLP-12 is expressed in DVA neurons. These experiments were performed at 30 and 120 min off food with EtOH treatment, (n = 10, N = 2 and F = 522, DF = 11). (C) Graph representing the number of body bends in mock and DVA ablated WT animals not treated with EtOH, (n = 10, N = 3 and F = 368, DF = 3). (D) Graph representing the amplitude of body bends in mock and DVA ablated WT animals not treated with EtOH, (n = 10, N = 3 and F = 192, DF = 3). (E) Graph representing the number of body bends quantitated from WT, *dop-2*, *ckr-2*, *DVAp::nlp-12* and *ckr-2*; *DVAp::nlp-12* treated with EtOH, (n = 10, N = 3 and F = 576, DF = 9). (F) Graph representing the amplitude of body bends quantitated from WT, *dop-2*, *ckr-2*, *DVAp::nlp-12* and *ckr-2*; *DVAp::nlp-12* animals treated with EtOH, (n = 10, N = 3 and F = 306, DF = 9). (G) Graph representing number of body bends quantitated from WT, *dop-2*, *ckr-1* and *ckr-1*; *dop-2* animals treated with EtOH, (n = 10, N = 3 and F = 1193, DF = 7). (H) Graph representing amplitude of body bends quantitated from WT, *dop-2*, *ckr-1* and *ckr-1*; *dop-2* animals treated with EtOH, (n = 10, N = 3 and F = 383, DF = 7). The same videos of moving animals were used to quantitate both number of body bends and the amplitude of body bends for each genotype. Error bars represent ±S.E.M., "n" represents the number of animals and "N" represents the number of replicates. The p-values were calculated using one-way ANOVA and Tukey-Kramer multiple comparison test; "*" indicates $p < 0.05$, "**" indicates $p < 0.01$, "***" indicates $p < 0.001$ and "ns" indicates not significant in all graphs. For both graphs the statistical comparison right above each genotype indicates a comparison with the respective WT control animals. Other statistical comparisons are indicated above or directly below lines indicating the genotypes that are being compared.
(TIF)

**S7 Fig. Mutants in the cholinergic pathway do not show a phenotype in the presence of EtOH.** (A) Graph indicating the number of body bends quantitated from WT, *cha-1* and *acr-16* mutants upon EtOH treatment (n = 10, N = 3 and F = 15.0, DF = 5). (B) Graph indicating amplitude of body bends quantitated from WT, *cha-1* and *acr-16* mutants upon EtOH treatment (n = 10, N = 3 and F = 10.8, DF = 5). (C) Graph indicating the number of body bends quantitated from WT, *dop-2*, *cha-1* and *myo-3p::acr-16* under with aldicarb (+A) and without aldicarb (-A) conditions (n = 10, N = 3 and F = 10.1, DF = 15). (D) Graph indicating amplitude of body bends quantitated from WT, *dop-2*, *cha-1* and *myo-3p::acr-16* under with aldicarb (+A) and without aldicarb (-A) conditions (n = 10, N = 3 and F = 2.30, DF = 15). The same videos of moving animals were used to quantitate both number of body bends and the amplitude of body bends for each genotype. Error bars represent ±S.E.M., "n" represents the number of animals and "N" represents the number of replicates. The p-values were calculated using one-way ANOVA and Tukey-Kramer multiple comparison test; "ns" indicates not significant in all graphs. For both graphs the statistical comparison right above each genotype indicates a comparison with the respective WT control. Other statistical comparisons are indicated above lines indicating the genotypes that are being compared.
(TIF)

**S1 Movie. WT (+EtOH) 120 min–Normal locomotory behaviour is observed in WT animals upon exposure to ethanol (EtOH) for 120 min.**
(MP4)

**S2 Movie. *dop-2* (+EtOH) 120 min–EIS locomotory behaviouris observed in *dop-2* mutant animals upon exposure to EtOH for 120 min.**
(MP4)

**S3 Movie. WT mock ablated (PDE) (+EtOH) 120 min–Normal locomotory behaviour is observed in WT mock ablated (PDE) animals upon exposure to EtOH for 120 min.**
(MP4)

**S4 Movie. WT PDE ablated (+EtOH) 120 min–Normal locomotory behaviour is observed in WT PDE ablated animals upon exposure to EtOH for 120 min.**
(MP4)

**S5 Movie. *dop-2* mock ablated (PDE) (+EtOH) 120 min–EIS locomotory behaviour observed in *dop-2* mock ablated (PDE) animals upon exposure to EtOH for 120 min.**
(MP4)

**S6 Movie. *dop-2* PDE ablated (+EtOH) 120 min–Normal locomotory behaviour was observed in *dop-2* PDE ablated animals upon exposure to EtOH for 120 min.**
(MP4)

**S7 Movie. WT+Dopamine (-EtOH) 120 min: Normal locomotory behaviour was observed in WT animals exposed to exogenous dopamine in the absence of EtOH at the 120 min time point.**
(MP4)

**S8 Movie. WT+Dopamine (+EtOH) 120 min: EIS like locomotory behaviour was observed in WT animals exposed to exogenous dopamine and treated with EtOH for 120.**
(MP4)

**S9 Movie. NLP-12 overexpression (+EtOH) 120 min- EIS like locomotory behaviour was observed in animals overexpressing NLP-12 in the presence of EtOH for 120 min.**
(MP4)

**S10 Movie. WT DVA ablated (-EtOH) 120 min–Defects in locomotory behaviour were observed in WT DVA ablated animals that were off food for 120 min.**
(MP4)

**S11 Movie. WT+aldicarb (+EtOH)–EIS like locomotory behaviour was observed in WT animals upon exposure to 1 mM aldicarb, followed by EtOH exposure for 120 min.** Please note that all movies involving exposure to EtOH were filmed at 120 minutes of exposure to 400 mM EtOH.
(MP4)

**S1 Text.** Includes Tables A, B and C that contain the complete list of strains, primers and plasmids used in this study.
(DOCX)

## Acknowledgments

The authors are especially grateful to Yogesh Dahiya for help with the FRAP experiment. We thank Randy Blakely, Rene Garcia and Ron Evans for reagents. A number of strains were

provided by the *Caenorhabditis* Genetics Centre, Minneapolis, USA. The authors thank Ankit Negi for routine help and the IISER Mohali Confocal facility for use of the confocal microscope.

## Author Contributions

**Conceptualization:** Pratima Pandey, Kavita Babu.

**Data curation:** Anuradha Singh.

**Formal analysis:** Pratima Pandey, Anuradha Singh, Harjot Kaur.

**Funding acquisition:** Kavita Babu.

**Investigation:** Pratima Pandey, Anuradha Singh.

**Methodology:** Pratima Pandey, Anuradha Singh, Harjot Kaur, Anindya Ghosh-Roy, Kavita Babu.

**Project administration:** Kavita Babu.

**Resources:** Anindya Ghosh-Roy, Kavita Babu.

**Supervision:** Pratima Pandey, Anindya Ghosh-Roy, Kavita Babu.

**Validation:** Pratima Pandey, Anuradha Singh.

**Visualization:** Pratima Pandey, Anuradha Singh.

**Writing – original draft:** Pratima Pandey, Anuradha Singh, Kavita Babu.

**Writing – review & editing:** Pratima Pandey, Anuradha Singh, Anindya Ghosh-Roy, Kavita Babu.

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
