## [Decision Letter · Decision Letter 0]

30 Jul 2020

Dear Dr Babu,

Thank you very much for submitting your Research Article entitled 'Increased dopaminergic neurotransmission results in ethanol dependent sedative behaviors in Caenorhabditis elegans' to PLOS Genetics. Your manuscript was fully evaluated at the editorial level and by independent peer reviewers. The reviewers appreciated the attention to an important problem, but raised some substantial concerns about the current manuscript. Based on the reviews, we will not be able to accept this version of the manuscript, but we would be willing to review again a much-revised version. We cannot, of course, promise publication at that time.

If you decide to revise the manuscript for further consideration at PLOS Genetics, please aim to resubmit within the next 60 days, unless it will take extra time to address the concerns of the reviewers, in which case we would appreciate an expected resubmission date by email to plosgenetics@plos.org.

[LINK]

We are sorry that we cannot be more positive about your manuscript at this stage. Please do not hesitate to contact us if you have any concerns or questions.

Yours sincerely,

Liliane Schoofs

Associate Editor

PLOS Genetics

Gregory P. Copenhaver

Editor-in-Chief

PLOS Genetics

Reviewer's Responses to Questions

**Comments to the Authors:**

Reviewer #1: In this study, Pandey et al. describe an ethanol-induced sedative behavior which occurs due to increased dopamine signaling by the PDE sensory neurons. They propose a model where dopamine activates the DVA neuron, promoting the release of NLP-12 neuropeptides which in turn activates cholinergic motor neurons. This causes a hypercontracted state and the authors suggest this is the cause of the sedation. This circuitry tested in this study was mostly previously described here: Davies et al 2003, Hawkings et al. 2015, Hu et al 2011). However, in the current study, EIS is being proposed as a novel behavior and the authors describe a new role for DOP-2.

I do have some significant concerns about the description of EIS, rationale and conclusions of the study.

1. The authors describe a “novel locomotory phenotype” when animals are exposed to 400mM EtOH. However, it appears that this behavior is largely artificial. It only occurs in the presence of non-physiological levels of dopamine (i.e. in the absence of dop-2 or upon the application of dopamine). It seems that a more accurate description would be that heightened dopamine inhibits recovery from acute ethanol exposure. In other words, the increased dopamine is just prolonging the activation of the circuit PDE -> DVA -> motor neurons which has been previously described (Hawkins et al. 2015).

2. The authors argue that this is a new behavior but the description of it is undeveloped. The only quantifiable components of this new behavior is 1 minute of body bending counts and a single measurement of body amplitude 120 minutes after ethanol exposure. Based on previous studies, ethanol is known to depress locomotion. In fact, in Davies 2003, they measure speed over time. It is unclear why this behavior is novel based on the way it is described. A temporal description of the dop-2 animal’s behavior compared to WT would make this more descriptive and useful. This would allow for other means of comparisons between genotypes and future studies. The authors explain what they qualitatively observed of the dop-2 animals between 0-120 minutes but do not show quantitative data. If EIS is novel, then a better description should be provided. Some of this is talked about in the methods section as well, but the data is not shown and it is unclear why it is in the methods section.

3. EIS reversibility is never quantified but just mentioned in the methods. Do the animals recover? If so, when? Are the animals just dying prematurely? Can they be rescued if removed from ethanol? Where is the data for this?

4. The conclusions on Lines 183 and 210 about EIS being a dop-2 dependent behavior are not correct. EIS only occurs in the absence of dop-2. It would be more accurate to suggest that EIS is induced by dopamine and a lack of dopamine signaling suppresses EIS.

5. Overall, the conclusions throughout the manuscript are hard to follow the way they are worded.

Some more minor concerns:

1. The authors provide a list of movies, however, the movies would benefit by having captions added to them.

2. Movie 5 – Animals look like rollers. Is the roller phenotype something that occurs with EIS and if so, can you discuss? The idea of them moving in circles is mentioned, is this related?

3. Supplementary figure 3 was skipped or numbering was off.

4. Line 92 – “…function through the G-protein signaling pathway.” This seems too vague, can you be more specific about which one and how? This seems relevant because the model predicts that these pathways are employed to increase neuropeptide release.

5. Line 99 – “….been associated with alcoholism.” It would be interesting to add a sentence that explains how they are associated, considering this is a major connection for this study.

6. Line 175 – It is unclear why these experiments were done. Is dopamine known to control pharyngeal pumping? Is dop-2 expressed in the pharyngeal circuitry? Also, the conclusions based on these experiments seem a bit broad. What about other muscles?

7. Line 184-185 (and 497) – standard C. elegans terminology is not used properly. dop-2p:dop-2:cfp

8. Line 201 – “defective signaling”? Isn’t it enhanced dopamine release?

9. Line 201 – The authors state they were unable to find a PDE specific promoter but then use a strain later with the asic-1 promoter. Rescue experiments could be done using this promoter.

10. Line 215 (Typo – DA neuron) and also is DOP-2 functioning through DVA? Or, is DOP-2 functioning to modulate dopamine from PDE to DVA? This should be reworded.

11. Lines 215-232 – It is very difficult to understand the rationale and conclusions here. For example, Line 227 – Removing dopamine suppresses EIS would seem more of an accurate summary of the results. Also, the conclusion on lines 231-232 are hard to follow. It is unclear what is meant by cat-2 functioning through dop-2?

12. Line 324 – more accurate to say nlp-12(OE)

13. Line 389 – typo “dopamine synthesis mutant”

14. Line 390 – more accurate to say acr-16(OE)

Reviewer #2: In this paper, Pandey et al identify a role of the dopamine receptor DOP-2 in regulating ethanol-dependent behaviours in C. elegans. The authors begin by testing mutants of the dopaminergic pathway in an established assay for ethanol-dependent behaviours, and find that dop-2 mutants display an ethanol-induced sedative (EIS) behaviour after prolonged ethanol exposure. After rescuing this phenotype by re-expressing dop-2 from its endogenous promoter, they ablate the posterior dopaminergic neuron PDE and find that it is required for the EIS behaviour of dop-2 mutants. Using genetic interventions, dopamine supplementation and FRAP as readout for synaptic vesicle fusion rate, they show that EtOH exposure results in increased dopamine neurotransmission from PDE synapses in dop-2 mutants. Further, the authors build on previous work, identifying DOP-1 and neuropeptidergic signalling from DVA as regulators of locomotion, to further characterize the mechanism that underpins ethanol-induced sedative behaviour in dop-2 mutants.

The experiments were performed thoroughly, and the evidence presented seems convincing (with a few exceptions, noted as major revision below). However, there are a number of issues throughout the manuscript that make the data difficult to interpret, and there are some aspects of the proposed mechanism that need to be clarified.

Major comments:

- The authors used laser ablation to determine the cellular focus of dop-2 in EIS behaviour, by ablating PDE neurons in wild type and dop-2 mutants. Although this experiment shows that PDE neurons are required for the EIS phenotype of dop-2 mutants, it does not proof that DOP-2 is required in PDE. An alternative explanation might be that dop-2 indirectly affects PDE function. The authors conclude that dop-2 functions in PDE neurons to regulate EIS behaviour, but additional evidence is needed to show that dop-2 indeed acts in PDE. A conditional system using promoters that uniquely overlap in PDE could be used when cell-specific promoters are not available.

- In the circuit model (Figure 7), the authors depict DOP-2 as an autoreceptor that regulates dopamine release in PDE, but they do not discuss the possibility that DOP-2 may act in additional neurons. DOP-2 is expressed in several other neurons where it may (also) indirectly affect PDE function. This should be clarified in the model and discussion, since the authors did not test a putative role of dop-2 in other neurons. Furthermore, additional evidence is needed to show that dop-2 is required in PDE for ethanol-dependent behaviours.

- Figures 2d-e and 4a-b show that PDE neurons are required for EIS behaviour in dop-2 mutants and have increased dopamine neurotransmission. Because dopamine is secreted, is it possible that other dopaminergic neurons show enhanced dopamine release and contribute to the EIS behaviour of dop-2 mutants as well?

- The figure panels do not show all statistical comparisons and often it is not clear which genotypes/conditions have been compared, making it difficult to interpret the data and assess whether the results support the main conclusions. All statistical comparisons should be included in the figure panels. For each p value, figure panels should clearly depict which genotypes/conditions were compared.

- The authors made several claims based on double mutant analyses (e.g. in Figure 3, 5 and 6), but only show statistical comparisons to wild type and not to the single mutants. Statistical comparisons of single and double mutants should be included to assess whether the dop-2 phenotype is mimicked or suppressed and whether genes function in the same genetic pathway or not.

- Also, in overexpression and supplementation experiments (Figures 3c-d, 5e-f, 6a-b, 6e-f), statistical comparisons are only shown for wild type making it difficult to assess if overexpression or supplementation suppresses or mimics the dop-2 phenotype. Statistical comparisons to other genotypes/conditions should be added to support the main conclusions.

- The authors state that loss of dop-1 function significantly decreases body bending (lines 288-290 and Figure 5a-b). However, this statement contradicts the data depicted in Figure S1-1 where dop-1 mutants did not show significantly altered body bending. The effect of dop-1 in ethanol-dependent behaviour should be clarified. If dop-1 mutants show locomotory defects after prolonged EtOH exposure, do these mutants have defects in the absence of EtOH?

- Similarly, the initial data on anterior body bending shows a defect in dop-2 mutants (e.g. in Figure 1f). However, this defect is not seen in some of the following experiments (e.g. in Figure 2b and 6d). Can the authors explain where this variability comes from?

- Based on DVA-specific rescue of dop-1, the authors conclude that the dop-2; dop-1 animals revert to the dop-2 like EIS behaviour (Figure 5a-b and lines 295-297). In Figure 5a-b, statistical comparisons between the transgenic rescue line and the double and single mutants should be included to support this statement.

- Overexpression of nlp-12 has previously been shown to affect basal locomotion and body bending in the absence of EtOH. These effects were found to be mediated by the receptors CKR-1 and CKR-2 (Ramachandran et al., 2020). The data presented in Figure 5g-h suggests that ckr-2 is involved in EIS behaviour of dop-2 mutants. Is this effect specific to CKR-2, or is CKR-1 involved as well?

- In Figures 6a-b and 6e-f, aldicarb treatment and acr-16 overexpression reduces body bending in wild-type animals exposed to EtOH. Are these effects specific to EtOH-exposure, or does aldicarb and acr-16 overexpression also affect body bending in the absence of EtOH? The latter is not shown but could imply that the observed effects are not specific to EIS behaviour. In Figure 6a-b, the cha-1 control strain shows a significant decrease in the frequency of body bending, unlike what the authors state in line 371. The corresponding control condition (cha-1 – EtOH), however, is not included in this figure. The authors should add the necessary controls in the absence of EtOH and aldicarb to figures 6a-b and 6e-f.

Minor comments:

- While the study appears to be sound, the authors should revise the language to improve readability and more clearly explain the rationale for the experimental work.

- The authors should carefully cross-check references in the main text and reference list. Some references mentioned in the main text seem to be missing in the reference list, e.g. Prescott and Kendler, 1999 and Schukit and Smith, 1996, Bettinger and Davies, 2014, etc.

- Several citations in the text are not formatted correctly, for example, line 346-347 “Shankar Ramachandran and Christopher M. Lambert, 2020), ...

- Individual data points in box plots are difficult to distinguish and would be clearer when plotted with smaller symbols.

- Throughout the manuscript, the main post-hoc test is annotated as Turkey-Kramer multiple comparison test. Do the authors refer to the Tukey-Kramer multiple comparison test?

- In Figures 2a-b, transgenes for rescue of dop-2 should be noted in italic, i.e. Pdop-2::dop-2. Likewise, in figure 5e-f, PDVA::nlp-12++ should be italic (not capitalized).

- Figure 6a-b, writing aldicarb instead of the abbreviation “A” would be clearer.

- Lines 44-46 and 62-66: Replace or remove “enhanced functioning of the DVA interneuron”, since DVA activity or output was not tested in dop-2 mutants. Similarly, the authors cannot claim that DVA releases NLP-12 resulting in the excitation of cholinergic motor neurons, because NLP-12 release and motor neuron activity was not tested after ethanol exposure.

- Line 86: change “synaptic molecules” because not all indicated examples have been located at synapses.

- Lines 140-143: Recent work has shown a role of DOP-2 autoreceptors as modulators of synaptic vesicle fusion in dopaminergic neurons (Formisano et al., 2020). Thus, dop-2 mutants have known defects associated with dopamine signalling. This work is relevant to the authors’ findings and should be referenced and discussed.

- Lines 146-154: The authors refer to a previous study by Davies and McIntire that studied the effects of 400 mM ethanol exposure on body bending. They state that similar phenotypes were observed for wild type animals in their study. However, in Figure 1, body bending was only quantified at 120 min after ethanol exposure. Short-term effects prior to this time point are not shown. The authors mention that dop-2 mutants behaved similar to wild type animals at earlier time points and are thus defective in recovery upon prolonged EtOH exposure. However, this data is not shown in the manuscript. The observation that dop-2 mutants show normal ethanol-induced behaviours upon short-term EtOH exposure is relevant to interpret the dop-2 phenotype (and to compare data in this study with previous work) and should be included in the manuscript.

- Line 158-161: The authors write that they “did not find any readily observable locomotory defects in other mutants in the dopamine pathway”. This contradicts the data in Figure S1-1, which shows significant phenotypes for cat-2 and dop-3 mutants. Mutants of slo-1 control animals also do not appear “insensitive to EtOH” based on the data in Figure S1-1.

- In Figure 1, the data in panels e and f cannot be interpreted without the results of the experiment in absence ethanol (depicted in panels g and h). Merge data in panels e and g and panels f and h, or include the figure panels without ethanol in e and f.

- In Figure 1, the data for wild-type animals (-EtOH) in panel c seems identical to the wild type data presented in panel g. Likewise, the data for wild-type animals (+ EtOH) in panel c seems identical to the data presented for wild type in panel e. Data for panels c, e and g should be merged or it should be indicated that the same wild type dataset was used in these figure panels. In addition, the figure legend of Figure 1 states that the same videos were used to analyse body bends/1 min (panels c/e/g) and amplitude (panels d/f/h), but the wild type data in panel d does not match the wild type data in panels f and h (as is the case for body bend/ 1 min). Is this an error, or were different videos used for analysing body bends/min and amplitude?

- Line 185: states that transgenic rescue lines for dop-2 expressed a Pdop-2::dop-2::cfp construct, whereas a Pdop-2::dop-2::mCherry construct is mentioned in the methods section.

- Figure 2d-e: Do PDE-ablated dop-2 mutants show a significantly higher frequency and amplitude in body bending when compared to mock-ablated dop-2 mutants?

- Lines 229-231: state that cat-2; dop-2 mutants show “similar behaviour as was seen in cat-2 mutants [...] These results indicated that cat-2 could be functioning through dop-2 [...]”. This statement is not supported by the data in Figure 3a-b, as cat-2 single mutants show increased body bending, whereas cat-2; dop-2 double mutants do not. The cat-2 mutation seems to suppress the dop-2 loss-of-function phenotype, although statistical comparisons to the single mutants should be included to support this conclusion. Furthermore, it is unclear from the presented data if cat-2 functions through (i.e. upstream) of dop-2. If dop-2 functions as an autoreceptor in PDE neurons, cat-2 may also be regulated by dop-2 function. This statement should be corrected, and statistical comparisons to the cat-2 and dop-2 single mutants should be included in Figure 3a-b.

- Line 253: Figure S3 is missing in the supplementary information.

- Figure 4b: Fluorescence recovery in wild type animals and dop-2 mutants (-EtOH) is fast at the start of the experiment and then quickly reaches a plateau around 20% recovery. This percentage is low compared to the work of Hardaway et al (2015) using the same reporter strain, where wild type animals showed recovery up to 80% in 2 minutes. Could the synapse have been damaged by photo-bleaching? Can the authors explain this difference in % recovery?

- Line 263-264: References supporting a role of D2-like receptors in DAT-1 localization are missing.

- The authors nicely show that loss of nlp-12 or ckr-2 suppresses the dop-2 phenotype after prolonged EtOH exposure, and that this phenotype is recapitulated by nlp-12 overexpression. Do nlp-12 and ckr-2 act in the same genetic pathway underlying EIS behaviour in dop-2 mutants? Testing nlp-12; ckr-2 double mutants or nlp-12(++); ckr-2 animals for EIS behaviour could further strengthen this notion and support the ligand-receptor interaction suggested in the circuit model (Figure 7).

- Is it known if EtOH exposure or increased dopamine release from PDE increases NLP-12 release from DVA? The authors include this statement in the final conclusion (lines 474-480) but did not quantify NLP-12 release in this study.

- Figure 6e-f: indicate the promoter used for overexpression of acr-16 in figure 6e-f.

- Line 419: states that this study shows for the first time the role of chronic EtOH treatment for extended periods of time up to 24 hours. All data presented in the study have been collected at 120 min of EtOH exposure which does not support this statement.

- Line 434-435: The conclusion that DAT-1 surface transport is dependent on DOP-2 is too strong. The authors only quantified DAT-1::GFP levels in dop-2 mutants but did not perform additional experiments, such as rescue of the DAT-1 transport phenotype, to show dependency on dop-2.

- Line 490: include the allele of ckr-2 used in this study (LSC32 is the strain name).

Reviewer #3: PLOS Genetics Review

Pandey et al

Increased dopaminergic neurotransmission results in ethanol dependent sedative behaviors in Caenorhabditis elegans

This is an interesting study describing a role for dopamine signalling in ethanol-induced behaviours using the model organism Caenorhabditis elegans. A role for the dopamine autoreceptor-encoding gene, dop-2, in the response to alcohol was identified. In addition, the study also reports a role for the posterior dopaminergic sensory neuron (PDE) in conveying this behaviour, and specifically shows that the exposure of dop-2 mutants to ethanol results in enhanced secretion of dopamine from PDE which stimulate the DVA interneuron to release a neuropeptide, NLP-12. The authors report that this then leads to excitation of the cholinergic motor neuron which impacts worm movement.

The authors employ a range of techniques, including behavioural bioassays, gene localisation tool, neuronal ablation and FRAP, to generate the data presented. The experiments appear to be robust and provide data that support the conclusions. This study provides interesting data for especially for researchers focussed on the dopaminergic pathway and the impact of alcohol on neurosignalling and behaviour. The manuscript is very well-written and the figures are clear.

Overall this paper presents useful data that should be communicated to researchers working in this area. In my opinion it would be suitable for publication in PLOS Genetics, and represents progress in the discipline.

General comments:

In my opinion no further experiments are essential for publication however a few additional experiments may provide interesting data that would aid further unravelling of the signaling system involved - the authors may wish to follow these up in future.

1. The authors describe the impact of ethanol on the body wall and pharyngeal muscle systems in the dop-2 mutant where they found an effect on body wall muscle (measuring locomotion) and no effect on pharyngeal muscle (measuring pumping), relative to WT worms. I wonder if the authors considered looking at the impact on reproductive muscle activity – using egg output as a measure of reproductive muscle function? This would have been interesting as the reproductive muscle is another major muscle system in nematodes, and the egg output assays are well-established for C. elegans.

2. The authors link the posterior DA neuron – PDE – to the phenotype observed on exposure of the dop-2 mutants to ethanol, and provide rationale for this approach, however there are additional DA neurons that have been identified in C. elegans (in addition to PDE) and I would have liked to have seem similar experiments for these neurons to determine if they also have a role in the observed effects? These cannot be ruled out.

3. Further to above, the authors focus on NLP-12 which is released from the DVA interneuron to enhance ACh release at neuromuscular junctions (previously established) – and they link this to the EIS phenotype (novel here). Does DVA contain any additional neuropeptides that may also be tentatively linked with the observed phenotype?

Minor comments:

I would like to make some minor comments on the manuscript which the authors may wish to consider.

Abstract and Author Summary:

It is my opinion that the abstract and author summary are well written and accurately reflect the content of the manuscript and the key outcomes of the research.

Introduction:

Generally, the introduction is quite well written and broadly covers the background detail required. Some minor considerations identified below.

Line 84

Replace ‘functions’ with ‘function’ or ‘function(s)’

Line 89-90

Replace……’which in turn induces’ with ‘which, in turn, induces’

Line 95

Replace: have also been identified, these receptors are thought to regulate the release of DA……with have also been identified; these receptors are thought to regulate the release of DA

Line 104 and again in Line 108, for example, species names should be written in full at the beginning of a sentence?

Line 109

Replace DA receptors, like their mammalian counterparts also belong to two subfamilies……with DA receptors, like their mammalian counterparts, also belong to two subfamilies…….

Line 110

In relation to the statement: EtOH shows its effect in a concentration dependent manner, acting as stimulant at lower concentration and depressant at higher concentration, please clarify if this is the effect on the receptor? Or another behavioural attribute? This is not clear and the statement appears to contradict the next sentence beginning…..’Studies in C. elegans have reported….’

The authors communicate that they developed an EtOH dependent assay but it is not clear if this is a novel assay? I understand that the phenotype observed on chronic exposure is novel?

Line 119

Replace ‘Mutants in the D2-like autoreceptor, dop-2 displayed a novel locomotory phenotype when exposed to 400 mM EtOH’ with ‘Mutants in the D2-like autoreceptor, dop-2, displayed a novel locomotory phenotype when exposed to 400 mM EtOH’

Results and Discussion:

The results are very well described and figures accurately reflect the content.

Line 147

Ethanol (EtOH) already introduced in line 145 so any abbreviations should come at the first introduction.

Line 158

‘We did not find any readily observable locomotory defects in other mutants in the dopamine pathway.’ – please clarify at this point which dopamine pathway mutants were examined.

Line 197 Bhattacharya et al should read Bhattacharya et al (2014) – also not sure that this should be communicated as a ‘recent’ publication…?

Line 419

Insert space between up and to

Line 466

Drosophila melanogaster should be abbreviated as spp. mentioned before.

Methods:

The methods provided do contain detail required to enable repeat of the experiments, and are in line with established protocols.

**Have all data underlying the figures and results presented in the manuscript been provided?**

Reviewer #1: Yes

Reviewer #2: None

Reviewer #3: Yes

PLOS authors have the option to publish the peer review history of their article (what does this mean?). If published, this will include your full peer review and any attached files.

Reviewer #1: No

Reviewer #2: No

Reviewer #3: No

---

## [Decision Letter · Decision Letter 1]

8 Dec 2020

Dear Dr Babu,

Thank you very much for submitting your Research Article entitled 'Increased dopaminergic neurotransmission results in ethanol dependent sedative behaviors in Caenorhabditis elegans' to PLOS Genetics.

The manuscript was fully evaluated at the editorial level and by independent peer reviewers. All reviewers appreciated the attention to an important topic but two reviewers still identified some concerns that we ask you address in a revised manuscript

We therefore ask you to modify the manuscript according to the review recommendations. Your revisions should address the specific points made by each reviewer.

[LINK]

Yours sincerely,

Liliane Schoofs

Associate Editor

PLOS Genetics

Gregory P. Copenhaver

Editor-in-Chief

PLOS Genetics

Reviewer's Responses to Questions

**Comments to the Authors:**

Reviewer #1: Overall, the authors have submitted an improved manuscript but some of my initial concerns remain unsatisfied. However, these are mostly minor concerns.

•The authors have addressed my concern about the novelty of this behavior and it is now clearer how this behavior occurs. I appreciate that they show a more detailed and temporal description of EIS in the manuscript. I still feel that the manuscript would benefit from a more detailed description, possible graphically in a main figure of the text. The additions to the supplement are nice but they seems slightly hidden. It is not clear to the average reader how EIS is unique from ethanol-induced paralysis. Couldn’t the back of the worm be paralyzed longer and that EIS is actually just a decoupling of the paralysis between the front and back? This was really the concern I had initially. It would help if the authors clarified this in the manuscript. Maybe ethanol-induced paralysis in dop-2 mutants was simply a paradigm for revealing a function for DOP-2.

•The authors have satisfactorily shown that EIS is reversible.

•The conclusions throughout are better worded but still hard to follow in a number of circumstances, which I have detailed below.

•The wording in lines 90-96 is still vague and grammatically incorrect. By saying dopamine receptors are GPCRs, then their participation with g-protein signaling would be implied. What G-proteins do they signal through based on work in other animals? Why not just say they signal through the subtype Galpha and be specific?

•With regards to my initial comment about the pharyngeal muscle, the authors have done additional experiments related to DOP-2’s role with egg-laying. Interestingly, egg-laying is also suppressed during their EIS behavior. This brings up the question, is egg-laying also restored in the rescued strains? Is this PDE-dependent or are other neurons sensitive to removing dop-2. This may be beyond the scope of the manuscript but would be a nice addition to the overall description of DOP-2’s function.

•Nomenclature for transgenics is still not used properly in the results, methods and figures. For example, lines 553, 559, 569. Using: dop-2p::dop2::cfp would be more consistent with the field. (See: https://wormbase.org/about/userguide/nomenclature#81al793kmj4hieg62d5fb0c--10)

•PDE-rescue could still be accomplished with an intersectional approach; however, the authors have explained that there is still a possibility that other neurons are involved. With this stipulation and the new experiments, they have satisfied my concern about PDE’s role here.

•Remove “hence” in line 255

•The conclusion in line 264-266 is unclear. Why unlikely? Can this be clarified?

•Lines 346-348. What do you mean by over-expression? Multi-copy array? Also, in the figure of which this refers to, it is unclear what PDVA::nlp-12 is referring to. Clarification is needed either here or in the methods, because the methods does not explain this either. PDVA is not the correct nomenclature.

•Line 373 – the nomenclature/grammar is not correct. Would be more accurate to say: “NLP-12 peptides signal through their receptor CKR-2.” The way it is phrased is suggesting you are talking about proteins. Also, there are multiple NLP-12 peptides (2). In fact, in this entire paragraph the nomenclature jumps back and forth.

•Again, the phrase DOP-2 functioning through CKR-1 is stated (line 380 and line 382). It would be more accurate to describe the mechanism. i.e. Mutations in dop-2 increase dopamine, which signals through DOP-1, increasing NLP-12, etc. It seems incorrect to just say DOP-2 functions through CKR-2. Same for lines 392,393.

•Line 395 – there are two distinct NLP-12 peptides. Would be more accurate to describe the model in those terms.

•Lines 426-428 are hard to follow. Also, NLP-12 (upper-case) refers to protein. So, “increased NLP-12 expression” is incorrect.

•Lines 440-441 – Not worded clearly. ACR-16(OE) is allowing for the EIS phenotype through the dopaminergic pathway? SO, ACR-16 is upstream of dopamine?

•Lines 444-445 – Same comment – conclusion is not worded clearly

•Missing the Ach label in fig 7a.

Reviewer #2: The authors have performed several additional experiments and clarified the main text and figures to address the reviewer comments. The additional analyses and discussion have substantially improved the quality of the manuscript, which I support for publication. I have no additional comments, except for the following typos:

Line 373: NLP-12 functions through its receptor...

Line 460: nucleus accumbens

Reviewer #3: The authors have effectively addressed all of the issues raised by this reviewer. The authors have performed significant additional experiments to address the comments of all reviewers and have enhanced the presentation of data and clarity in their interpretation of findings. In general, the readability of the manuscript has also improved.

**Have all data underlying the figures and results presented in the manuscript been provided?**

Reviewer #1: Yes

Reviewer #2: Yes

Reviewer #3: Yes

PLOS authors have the option to publish the peer review history of their article (what does this mean?). If published, this will include your full peer review and any attached files.

Reviewer #1: No

Reviewer #2: No

Reviewer #3: No

---

## [Editor Report · Decision Letter 2]

6 Jan 2021

Dear Dr Babu,

We are pleased to inform you that your manuscript entitled "Increased dopaminergic neurotransmission results in ethanol dependent sedative behaviors in Caenorhabditis elegans" has been editorially accepted for publication in PLOS Genetics. Congratulations!

Yours sincerely,

Liliane Schoofs

Associate Editor

PLOS Genetics

Gregory P. Copenhaver

Editor-in-Chief

PLOS Genetics

Comments from the reviewers (if applicable):

**Data Deposition**

http://datadryad.org/submit?journalID=pgenetics&manu=PGENETICS-D-20-00998R2

**Press Queries**

---

## [Editor Report · Acceptance letter]

25 Jan 2021

PGENETICS-D-20-00998R2 

Increased dopaminergic neurotransmission results in ethanol dependent sedative behaviors in Caenorhabditis elegans 

Dear Dr Babu, 

We are pleased to inform you that your manuscript entitled "Increased dopaminergic neurotransmission results in ethanol dependent sedative behaviors in Caenorhabditis elegans" has been formally accepted for publication in PLOS Genetics! Your manuscript is now with our production department and you will be notified of the publication date in due course.

With kind regards,

Alice Ellingham

PLOS Genetics

On behalf of:
